# Beyond single neurons: population response geometry in digital twins of mouse visual cortex

**Dario Liscai**[1,*]**, Emanuele Luconi**[1,*]**, Alessandro Marin Vargas**[2]**, Alessandro Sanzeni**[1,†]

[1]Bocconi University, Italy
[2]École Polytechnique Fédérale de Lausanne, Switzerland
[*]These authors contributed equally to this work, [†]alessandro.sanzeni@unibocconi.it

## Abstract

Hierarchical visual processing is essential for cognitive functions like object recognition and spatial localization. Traditional studies of the neural basis of these computations have focused on single-neuron activity, but recent advances in large-scale neural recordings emphasize the growing need to understand computations at the population level. Digital twins–computational models trained on neural data–have successfully replicated single-neuron behavior, but their effectiveness in capturing the joint activity of neurons remains unclear. In this study, we investigate how well digital twins describe population responses in mouse visual cortex. We show that these models fail to accurately represent the geometry of population activity, particularly its differentiability and how this geometry evolves across the visual hierarchy. To address this, we explore how dataset, network architecture, loss function, and training method affect the ability of digital twins to recapitulate population properties. We demonstrate that improving model alignment with experiments requires training strategies that enhance robustness and generalization, reflecting principles observed in biological systems. These findings underscore the need to evaluate digital twins from multiple perspectives, identify key areas for refinement, and establish a foundation for using these models to explore neural computations at the population level.

## 1 Introduction

Our ability to perform complex tasks, such as object recognition and spatial localization, relies on hierarchical processing of visual information in the brain. Studies using single-neuron recordings have found a progressive transformation of visual representations: from simple feature detection in early sensory areas to complex object representations in higher visual areas (Felleman & Van Essen, 1991; Hubel & Wiesel, 1968). Advances in large-scale recordings have shifted the focus from single-neuron activity to population-level responses. These provide a more comprehensive view of the neural processes supporting cognitive functions and have been suggested to be the essential unit of computation (Chung & Abbott, 2021; Vyas et al., 2020; Saxena & Cunningham, 2019).

A major challenge in studying the neural basis of complex computations, such as object classification, is understanding how the joint activity of neurons represent stimuli and how these representations are transformed across brain areas. Limitations in consistently recording from large numbers of neurons under diverse conditions constitute a significant barrier to addressing this challenge. Artificial neural networks trained on large-scale neural recordings (Fig. 1A), also referred as *digital twins*, have recently emerged as powerful tools for bridging this gap (Lurz et al., 2020; Safarani et al., 2021; Wang et al., 2023). By learning to predict neural responses, these models offer the opportunity for extensive in silico experiments, which would be impractical in live animals, such as identifying the most excitatory stimuli (Walker et al., 2019; Bashivan et al., 2019; Pierzchlewicz et al., 2024) or mapping the landscape of relevant features across neuronal populations (Tong et al.,

2023). Despite their success at the single-neuron level, the potential of digital twins for studying population-level neural responses remains underexplored.

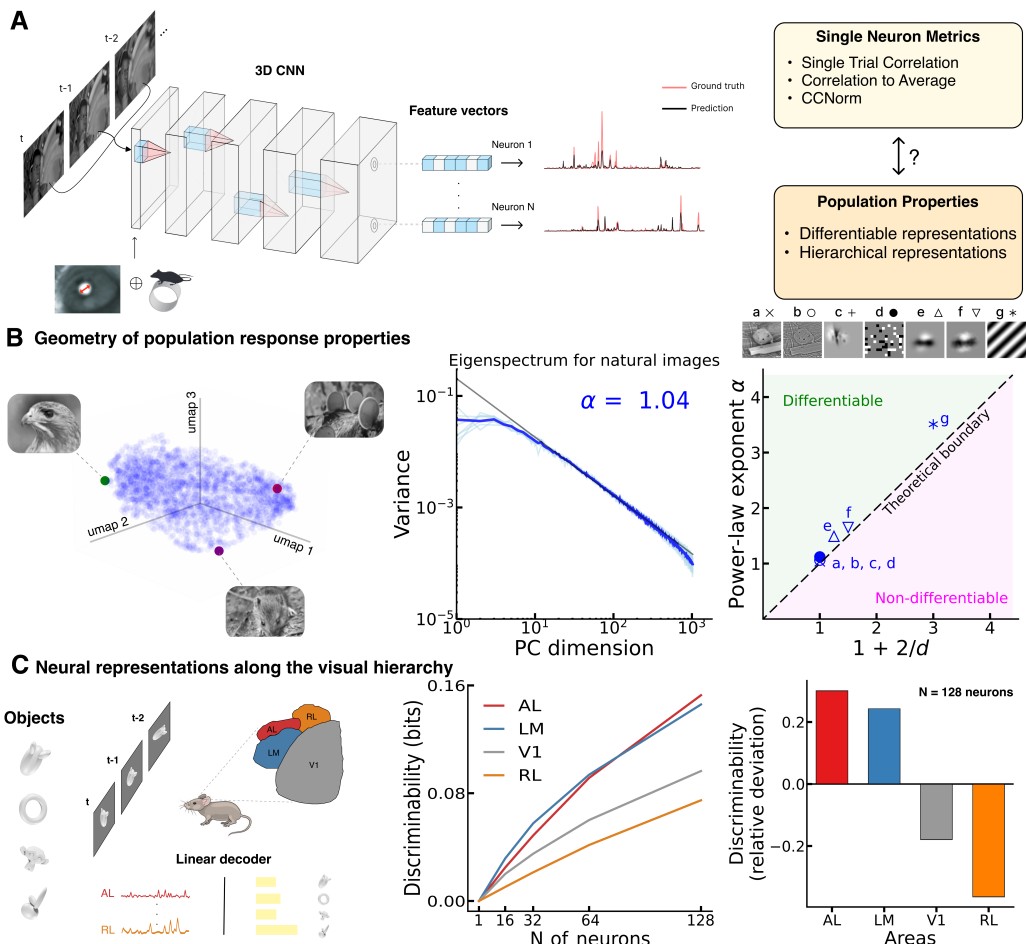

Figure 1: **Investigating population geometry of mouse visual cortex with digital twins** (**A**) Schematic of a digital twin used to predict single neuron responses. We study how well these models capture the neural response geometry and its variation across cortical areas. (**B**) In Stringer et al. (Stringer et al., 2019a), mice were exposed to thousands of natural images and parametric stimuli, grouped into ensembles of varying dimensions $d$, while V1 neuronal activity was recorded simultaneously. The collective neuronal activity corresponding to each stimulus type formed a manifold (left), whose structure was analyzed using the eigenspectrum of the covariance matrix of neural activity, which exhibited a power-law decay (center; lighter and darker blue lines represent individual sessions and their average, respectively). The decay exponent $\alpha$ depended on $d$ and slightly exceeded $1 + 2/d$ (right), suggesting that the neural representations are as high-dimensional as possible while still being differentiable. (**C**) In Froudarakis et al. (Froudarakis et al., 2020), mice were presented with objects undergoing identity-preserving transformations (rotation, scaling, translation, and variations in lighting) while simultaneous recordings were made from thousands of neurons (left). The experiment revealed a hierarchy of linearly decodable object information across different visual areas (center and right).

In the visual cortex, neural population responses to visual stimuli form manifolds, whose structure determines how efficiently downstream neural circuits can differentiate between stimuli, generalize across appearance variations, and maintain robustness against noise (DiCarlo et al., 2012; Chung et al., 2018; Chung & Abbott, 2021; Li et al., 2024). Accurately capturing the structure of these population-level responses, both within individual brain areas and as it evolves across the visual hierarchy, is important for digital twin models aiming to provide insights into neural computations. In this study, we evaluate the ability of digital twins to replicate two landmark findings in the mouse visual cortex hierarchy. First, we investigate the differentiability of population responses in primary visual cortex (V1). Stringer et al. (Stringer et al., 2019a) demonstrated that V1 representations exhibit a high-dimensional geometry that remains differentiable (Fig.1B), enabling robustness to noise

and small variations in stimuli. They achieved this by exposing mice to ensembles of visual stimuli (e.g., natural images and parametric stimuli) grouped by dimension $d$ and analyzing the resulting neural activity manifolds using the eigenspectrum of the covariance matrix. The spectrum followed a power-law decay, with an exponent $\alpha$ dependent on $d$ and slightly exceeding $1 + 2/d$. Stringer et al. (Stringer et al., 2019a) emphasized that a spectrum with too slow a decay is pathological, as it requires an impractically large number of neurons to reconstruct stimuli and makes neural responses extremely sensitive to small stimulus changes—both of which hinder learning to generalize. Mathematically, they showed through functional analysis that the decay exponent $\alpha$ reflects the geometry of neural representations. Specifically, the dimensionality of representations increases with $\alpha$, and their differentiability depends on key thresholds: representations with $\alpha < 1$ are discontinuous, those with $1 \leq \alpha < 1 + 2/d$ are continuous but not differentiable, and those with $\alpha > 1 + 2/d$ are differentiable. These results are based on the relationship between the eigenspectrum and the similarity of arbitrary neural response vectors. If the spectrum decays too slowly, the trace of the covariance matrix (linked to vector similarity) diverges, causing instability in neural representations.

Second, we investigate how neural representations transform across cortical areas. Experiments have revealed signatures of hierarchical information processing in mouse visual cortex, at both the anatomical (Wang et al., 2012; D'Souza et al., 2022) and functional (Siegle et al., 2021; Froudarakis et al., 2020; Hoeller et al., 2024) level. We focus on the experiment by Froudarakis et al. (Froudarakis et al., 2020), which demonstrated that neural representations evolve to support invariant object recognition (Fig. 1C). They achieved this by recording neural activity in various areas while presenting objects undergoing identity-preserving transformations, such as rotation, scaling, and translation, and evaluating the discriminability of these representations across different areas.

We find that state-of-the-art digital twins struggle to replicate these experimental results. To tackle this issue, we examine the impact of different dataset compositions, network architectures, loss functions, and regularization techniques on the models' capacity to capture population-level responses. We show that adding regularization during training improves the model's ability to learn neural representations that match the experimental population response. Our findings underscore the need for digital twin models to move beyond single-neuron predictions and push toward modelling the complexities of neural population activity and its hierarchical transformations within the visual system.

## 2 RELATED WORKS

**Encoding models of the visual cortex.** Encoding models of the visual system aim to characterize how sensory stimuli are transformed into neural responses. These models have evolved from linear-nonlinear Poisson (LNP) models (Simoncelli et al., 2004) to deep artificial neural networks (ANNs), which are now considered digital twins of the brain's visual processing pathway. Two primary approaches dominate: task-driven models, which optimize ANNs to solve specific computational tasks like object recognition (Yamins et al., 2014; Bakhtiari et al., 2021; Kubilius et al., 2019; Zhuang et al., 2021; Pogoncheff et al., 2023), and data-driven models, which are trained directly on neural data to replicate observed responses (Klindt et al., 2017; Cadena et al., 2019; Lurz et al., 2020; Safarani et al., 2021; Willeke et al., 2022; Li et al., 2023; Wang et al., 2023). Recently, new models have been developed to address the complexity of dynamic stimuli, like videos. From the best models of the Dynamic Sensorium Competition (Turishcheva et al., 2024) to multi-modal models that combine visual images with behavioral variables (Xu et al., 2024), these models have achieved great prediction accuracy. Despite these models' success in capturing single-neuron responses, they struggle to develop hierarchical representations (St-Yves et al., 2023; Dyballa et al., 2024) and we show they also fail to represent the complex structure of neural population responses.

**Single-neuron and population-level analysis.** Applications of neural network models in visual neuroscience have largely focused on single-neuron properties, investigating how individual neurons respond to stimuli. Previous works have examined feature selectivity for object recognition and texture perception (Walker et al., 2019), the columnar organization of selectivity in V4 (Willeke et al., 2023), and the encoding of phase-invariant or fixed spatial patterns in V1 (Ding et al., 2023; Ustyuzhaninov et al., 2022). While these analyses have provided valuable insights, they fall short in capturing the population structure of neural responses. Recent work has started to address this limitation. Pogoncheff et al. (Pogoncheff et al., 2023) explored the relationship between V1 predictability and population-level properties. Similarly, Margalit et al. (Margalit et al., 2024) examined

population-level topological properties across early (V1) and higher-order (VTC) visual areas. We continue this shift towards population-level analysis by using digital twins to investigate the geometrical properties of neural population responses, with a focus on differentiability and discriminability.

**Regularization strategies.** To bridge the gap between observed neural activity and artificial models, various studies have introduced regularization techniques, often drawing inspiration from principles like the efficient coding hypothesis (Barlow et al., 1961). Sparsity constraints, for example, helped neural networks to develop Gabor-like receptive fields, mimicking the tuning properties of early visual neurons (Olshausen & Field, 1996; Bell & Sejnowski, 1997). These regularizations can be applied during model training, e.g. adversarial training to improve robustness (Guo et al., 2022) or through image augmentation techniques that enhance model generalization (Shorten & Khoshgoftaar, 2019). More recent approaches apply architectural constraints, e.g. functional or topological constraints, to better align artificial models with biological data, enhancing their interpretability and functional similarity (Pogoncheff et al., 2023; Margalit et al., 2024). Our study builds on these approaches by demonstrating that introducing regularization strategies significantly enhance the models' ability to learn representations aligning with those observed in biological systems.

## 3 DIGITAL TWIN OF THE MOUSE VISUAL CORTEX

To evaluate the capability of digital twins of the mouse visual cortex in capturing both single-neuron and population-level response properties, we used the baseline model of the SENSORIUM competition (Turishcheva et al., 2023), whose architecture is structured into three components: a core module, a readout module, and a shifter module. The core is composed of 4 layers of 3D Convolutional Neural Network (CNN), inspired by (Höfling et al., 2022), which encodes the spatiotemporal features of visual stimuli and behaviors into a latent representation (see Supp. A.1). The readout, taken from (Lurz et al., 2020), transforms the core representation into individual neuronal activity. Its functioning is divided into two components: a spatial mask, which determines the neurons' receptive field positions (i.e., where they "look" in the image), and a feature weight, which linearly combines features from the core's final layer to generate neural responses frame-by-frame. The shifter takes the pupil center coordinates as input and produces a "gaze-shift" that is linearly applied to the spatial positioning of all neurons' receptive fields. The output is then passed through an activation function (ELU +1) to generate the final neuronal responses. We introduced a few minor modifications to the original architecture (see list in A.1), which we found to facilitate training without modifying performance (see Table 1).

We trained the model using the MICrONS dataset (Consortium et al., 2021), which includes more than 75,000 neurons spanning multiple areas of the mouse visual cortex (V1, LM, AL, RL) collected in 14 recording sessions for a single mice. The network was trained to minimize the Poisson loss between predicted and recorded responses (see Supp. A.1). Throughout this paper, we refer to this trained model as the benchmark model. The model demonstrated strong performance in single-neuron predictions, achieving a median normalized correlation coefficient ($CC_{\text{norm}}$) of 0.47–0.58 across sessions. This result is comparable to the performance reported by (Wang et al., 2023) (see Table 1), which, to the best of our knowledge, is the only other model trained end-to-end on the MICrONS dataset available in literature. Performance slightly varied across visual areas (Supp. Fig. 1), with the highest prediction accuracy observed in V1 (mean corr-to-average 0.305), followed by LM (0.28), RL (0.255) and AL (0.253). These performance differences across areas may reflect a higher number of neurons in certain regions, which could significantly impact the loss function (83,222 units in V1, 14,817 in LM, 12,599 in RL and 4,734 in AL across all sessions); alternatively, differences in single-neuron response reliability (6.77% in V1, 6.37% in LM, 5.08% and 5.03% in AL calculated as Fraction of Explainable Variance over repeated stimuli) might bias the model to learn one area better than another, with both factors potentially coexisting.

## 4 DIGITAL TWINS FAIL TO CAPTURE THE STRUCTURE OF POPULATION RESPONSES IN MOUSE VISUAL AREAS

Traditional metrics for training and evaluating digital twins, such as Poisson loss and single-neuron correlation, focus on individual neuron predictions and overlook population-level structures arising from neural interactions. This gap raises questions about whether these models can capture

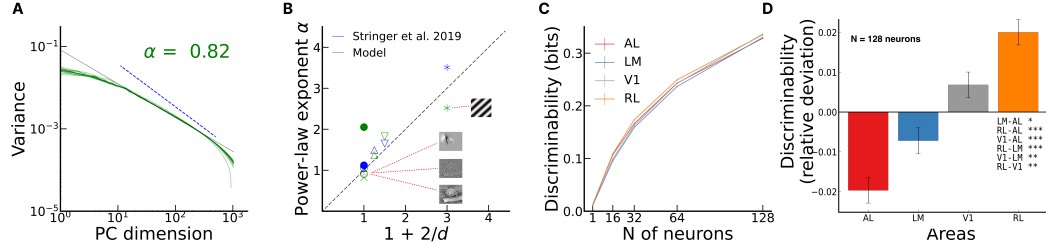

Figure 2: **Population geometry in a 3D convolution model** (**A**) Eigenspectrum of the covariance matrix obtained from our benchmark model, simulating the experiment by Stringer et al. (Stringer et al., 2019a) with natural images (lighter green lines are individual sessions; the darker green line is the average). The eigenspectrum exhibits a power-law decay (best-fit exponent $\alpha = 0.82$) significantly slower than that observed in experimental data from Stringer et al. (Stringer et al., 2019a) (blue, exponent $\alpha = 1.04$). (**B**) Power-law exponents for the model (green) and experimental data from Stringer et al. (Stringer et al., 2019a) (blue), calculated across different types of visual stimuli with varying dimensionality. The digital twin's representation aligns with experimental observations, showing higher dimensionality with increased stimulus complexity. However, it exhibits a slower decay for natural, whitened, spatially localized images and gratings, indicating that the learned representation of these stimuli is not differentiable. (**C**) Hierarchy of neural representations obtained from the benchmark model when simulating the experiment of Froudarakis et al. (Froudarakis et al., 2020). We evaluated the ability of a linear classifier to predict object identity across conditions using neural representations from different areas generated in the model. Results represent the aggregation of three independent model runs to ensure robustness and consistency. (**D**) Relative discriminability across areas computed in the model, as in Fig. 1C. The model fails to capture the differences in discriminability observed experimentally across visual areas. Here and in what follows: error bars represent the standard error of the mean (s.e.m.); stars indicate significance levels from the Mann–Whitney U test ($\ast$, $\ast\ast$, $\ast\ast\ast$ indicate p-val$< 0.05, 0.01, 0.001$).

key properties of population response structure observed experimentally in visual cortex (Stringer et al., 2019a; Froudarakis et al., 2020). To address this, we replicated these experiments with our benchmark model, testing its ability to capture the observed population-level properties.

To analyze if digital twins capture the geometry of the neural representation of V1, we simulated the experiment of Stringer et al. (Stringer et al., 2019a). We presented ensembles of images with varying dimensionalities to the model, extracted predicted responses from neurons in layer 2/3 of V1, and measured the covariance between neural responses across images, mimicking the experimental procedure (Fig. 1B and Supp. A.2). Despite the visual stimuli from Stringer et al. being substantially different from those used in training, the model did not exhibit any pathological response (Supp. Fig. 1), suggesting that it can generalize to these experimental protocols. Consistent with the experimental data, the spectrum of the covariance matrix between units in the model demonstrated a power-law behavior for natural images (Fig. 2A) and for other types of visual stimuli investigated (Supp. Fig. 1). Furthermore, the model recapitulates the qualitative dependency on stimulus dimensionality observed experimentally, showing a smaller power-law exponent $\alpha$ for higher-dimensional stimuli (Fig. 2B). Nevertheless, we found significant discrepancies between the model and the experimental results. Stringer et al. showed that the power law exponent $\alpha$ of the eigenspectrum

Table 1: Comparison of model single-neuron prediction performance (definitions in Supp. A.1) on the MICrONS (Consortium et al., 2021) and SENSORIUM (Turishcheva et al., 2023) datasets. For SENSORIUM, results for (Turishcheva et al., 2023) are relative to the live test set, whereas ViV1T and ours refer to the validation set.

|  | Dataset (cortical areas) | $CC_{\text{norm}}$ | Corr | Corr-to-average |
|---|---|---|---|---|
| Wang et al. (2023) | MICrONS (V1,LM,RL,AL) | 0.48-0.65 | - | - |
| **Our benchmark model** | MICrONS (V1,LM,RL,AL) | 0.47-0.58 | 0.168 | 0.294 |
| Turishcheva et al. (2023) | SENSORIUM (V1) | - | 0.229 | 0.416 |
| ViV1T (Turishcheva et al., 2024) | SENSORIUM (V1) | - | 0.261 | 0.478 |
| **Our benchmark model** | SENSORIUM (V1) | - | 0.247 | 0.452 |

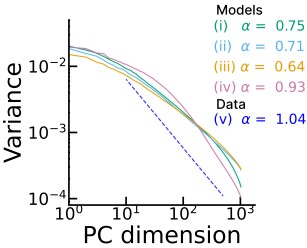

Figure 3: **Non-differentiable geometry of V1 population responses across network models.** Eigenspectrum of the covariance matrix for natural stimuli in: (i) model trained on reliable neurons, (ii) model trained on the SENSORIUM dataset, (iii) ViV1T, (iv) model trained on correlations, and (v) experimentally observed power-law decay. Eigenspectra for other stimuli are shown in Supp. Fig. 2. Despite providing better descriptions of either single neuron responses or the correlations between them compared to our benchmark model, none of these models exhibited a differentiable representation.

encapsulates key features of the geometry of neural representations. Smaller $\alpha$ indicates higher-dimensional representations; $\alpha < 1$ means the representation is not continuous, and $\alpha < 1 + 2/d$ means it is not differentiable. In mice, $\alpha$ was found to be slightly above $\alpha = 1 + 2/d$, suggesting a nearly maximal dimensionality while maintaining differentiability. This allows the network to represent a wide range of stimuli smoothly, aiding downstream decoders in generalizing from small stimulus changes. In our benchmark model, for different stimuli (natural images, spatially localized stimuli, whitened stimuli, and gratings), we found a power law exponent $\alpha$ below the critical line $1 + 2/d$ (Fig. 2B). These findings indicate that, unlike the representation in the mouse V1, the representation learned by the digital twin is not differentiable and, for natural, whitened and spatially localized images, is not continuous.

To assess whether digital twins can account for hierarchy in discriminability observed across visual areas, we simulated the experiment by Froudarakis et al. (Froudarakis et al., 2020). In their study, mice were shown movies of objects undergoing identity-preserving transformations, including rotation, scaling, translation, and variations in lighting. We recreated these movies, fed them into our benchmark model and extracted the predicted neural responses from different visual areas (See Supp. A.3). Despite the significant differences between these stimuli and the training data, the model produced plausible responses (Supp. Fig. 1). We then used these extracted responses to train cross-validated logistic regression models for object identification (Fig. 2C,D and Supp. A.3). The model showed notable quantitative discrepancies, with discriminability values higher than those reported experimentally and less pronounced differences between visual areas (Fig. 2C). While these discrepancies are expected due to the absence of spiking noise, reduced variability, and differences in visual stimuli, changes in discriminability across areas (Fig. 2D) revealed a misrepresentation of the hierarchy observed in (Froudarakis et al., 2020), with uniform object identity discriminability across the four brain areas. This qualitative discrepancy indicates that the model fails to capture the hierarchy in discriminability observed across visual areas of the mouse cortex.

These results show that while our model captures single-neuron responses, it fails to represent the population-level structure and hierarchical transformations. Capturing the hierarchy of discriminability across areas likely requires first understanding the representation geometry within an area, as this determines decoder performance (Chung et al., 2018; Chung & Abbott, 2021; Li et al., 2024). Thus, in what follows, we will first modify the model to capture the geometry of population responses in V1, then examine how these changes affect the hierarchy of discriminability across areas.

## 5    TOWARDS CAPTURING V1 REPRESENTATION GEOMETRY

To develop digital twins of mouse V1 that reflect neural response geometry, we first assessed the robustness of our findings, by evaluating how variations in the dataset, training procedure or network architectures may impact the ability to capture the geometry of V1 responses. After finding these methods inadequate, we show that specific regularization techniques push the model toward capturing representations consistent with those of mouse V1.

### 5.1    DIGITAL TWINS CONSISTENTLY FAIL TO REPLICATE DIFFERENTIABILITY OF NEURAL REPRESENTATIONS

We first examined whether improving model performance could lead to improvements in capturing the geometry of population responses. Experimental studies show that spontaneous activity overlaps with visual responses, causing variability in neuron responses to repeated stimuli (Niell & Stryker,

2010; Stringer et al., 2019b; Avitan & Stringer, 2022). Although the model partially accounts for this variability by including running and pupil size as inputs, it remains insufficient due to: (1) the influence of additional behavioral variables on spontaneous activity (Stringer et al., 2019b), and (2) the fact that not all spontaneous activity is behavior-driven (Stringer et al., 2019b). These limitations constrain the model's accuracy in predicting single-neuron responses. We trained our model to replicate exclusively the activity of neurons with high reliability to repeated videos (see Supp. A.4). As expected, model performance improved for both single trials and trial-averaged responses, with correlation increasing by 19.0% and 36.7%, respectively. Nevertheless, the spectral properties remained unchanged (Fig. 3i and Supp. Fig. 2), suggesting that while individual response predictions improved, the underlying population response geometry was not affected.

We then considered whether the characteristics of the training dataset influenced our findings. To investigate this, we trained our model using the SENSORIUM competition dataset (Turishcheva et al., 2023) (see Supp. A.4). The model trained on SENSORIUM significantly outperformed the MICrONS-trained model, showing improvements in correlations between single-neuron responses and model predictions by a factor of 47.0% in single trials and 53.7% in trial averages. Despite this improvement, the model's ability to capture the power-law exponents observed in experimental data did not improve, either for natural images (Fig. 3ii) or other tested stimuli (Supp. Fig. 2). As a result, the representation remained non-differentiable thereby discarding the training dataset as a possible factor to influence the population geometry.

We then examined whether the model's architecture affected our results. CNNs have historically been prominent in computer vision, but recent advances have shown that transformers are now achieving competitive and sometimes superior performance in various tasks (Khan et al., 2022; Azabou et al., 2024; Antoniades et al., 2023). Motivated by this, we employed ViV1T, a transformer-based digital twin trained on the SENSORIUM dataset (Turishcheva et al., 2024), to replicate the experiment from Stringer et al. (Stringer et al., 2019a). Our results show that ViV1T, like previous models, fails to capture the geometric properties of mouse V1, with representations remaining non-differentiable despite improved accuracy (Fig. 3iii and Supp. Fig. 2).

Finally, we explored whether modifying the target of the loss function could improve the model's ability to capture the covariance between neurons. Previous models were optimized to predict individual neuron responses, which prioritized single-neuron accuracy over population structure. To address this, we developed a new training scheme where the loss function is based uniquely on the correlations between neurons. The trained network significantly outperformed our benchmark model in describing correlations between neurons (root mean squared error on correlations decreased by 41.76%). Unsurprisingly, this improvement came at the expense of the model's ability to capture single-neuron responses, which was significantly decreased. We replicated the experiment by Stringer et al. (Stringer et al., 2019a) and analyzed the covariance matrix spectrum (Fig.3iv and Supp. Fig. 2). Unlike previous approaches, this model displayed power-law behavior with a significantly larger $\alpha$ than the benchmark model (0.93 for natural stimuli compared to 0.82 of the original model). However, the decay was still slower than observed experimentally, and the resulting representation remained non-differentiable. We expanded this approach with multi-objective optimization to balance single-neuron responses and population correlations (Supp. Fig. 8). We found no evidence of a synergistic effect: gradually increasing the relative weight of the Poisson loss led to models better capturing single-neuron responses but at the expense of the differentiability of population representations. Additionally, we explored whether task-driven modeling could yield more differentiable representations. Specifically, we fine-tuned a linear readout to predict neural responses after each of the first three layer block of a ResNet50 with frozen weights trained on object recognition. Consistent with the data-driven models, the task-driven model also produced a non-differentiable representation (Supp. Fig. 6), suggesting that the chosen objective functions lack the inductive bias necessary for developing differentiable representations.

## 5.2 Training regularization facilitates high-dimensional, differentiable representations in digital twins

Our analyses reveal that digital twins are prone to learning neural representations that are not differentiable. We hypothesize that introducing strategies to enhance robustness in neural responses could improve differentiability. Specifically, we reasoned that training the model with pairs of images, where neural responses are kept fixed while the images are slightly perturbed, would encourage the

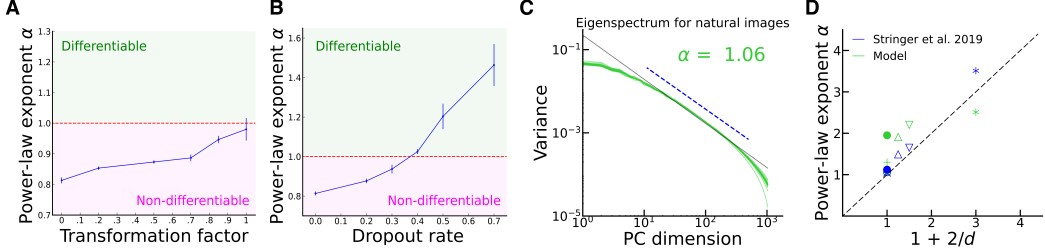

Figure 4: **Effects of training regularization in digital twin models** (**A**) Power-law exponent $\alpha$ measured in digital twins simulating responses to natural images (from (Stringer et al., 2019a)) with respect to the amplitude of image transformation used during training (see Supp. A.1). (**B**) Power-law exponent $\alpha$ of natural images with respect to the dropout rate. Digital twins with a dropout rate greater than 0.4 exhibit an exponent larger than 1, indicating a differentiable representation of natural images. (**C**) Eigenspectrum of the covariance matrix in response to natural images with a dropout rate of 0.4. The model exhibits a power-law decay of $\alpha = 1.06$, closely matching the experimental data of $\alpha = 1.04$. (**D**) Power-law exponent as a function of stimulus dimensionality for a dropout rate of 0.4. For all types of visual stimuli, except gratings, the model exhibits a differentiable geometry of the neural representation, consistent with experimental data.

model to learn representations in which small changes in the input produce minor changes in the neural responses. This is effectively a way to ensure that the representations are differentiable.

To test our hypothesis, we trained the model using stimulus augmentation, applying random transformations (e.g., translation, rotation, rescaling) to the input images. These transformations were chosen for their relevance to the differentiability of neural representations, simulating the way animals perceive stimuli from various angles and distances during movement. The model exhibited a modest decline in performance when predicting single-neuron responses as transformation amplitude increased (Supp. Fig. 3), while the covariance matrix retained a power-law spectrum (Supp. Fig. 3). Larger transformations during training led to more robust neural representations, with the power-law exponent of the spectrum approaching that of differentiable representations as the transformation amplitude increased (Fig. 4A).

Another potential approach to enhancing the robustness of neural representations is to introduce perturbations directly within the network itself. One widely used machine learning technique that adopts this strategy is dropout (Srivastava et al., 2014). By randomly disabling units during training, dropout effectively reduces overfitting and encourages the model to generalize better across unseen data. We hypothesized that this method could also improve the model's ability to capture the underlying structure of neural responses. To test this, we applied dropout across layers of the 3D CNN during training and analyzed its impact on the model's representation of neural activity. While dropout reduced performance as the dropout rate increased (Supp. Fig. 4), the model still accurately predicted single-neuron responses and maintained a power-law spectrum for the covariance matrix (Supp. Fig. 4). Gradually increasing the dropout rate significantly modified the power spectrum across all stimulus types, achieving differentiable representations of natural images for dropout rate $\geq 0.4$ (Fig. 4B,C). Differentiable representations, consistent with those observed experimentally, emerged for all visual stimuli investigated, except for grating stimuli (Fig. 4D and Supp. Fig. 4); we will discuss possible explanations for this exception in the discussion section.

# 6 IMPACT OF REPRESENTATION GEOMETRY ON THE HIERARCHY LEARNED BY DIGITAL TWINS

The analysis of Fig. 2C,D shows that a digital twin trained to reproduce single-neuron responses fails to capture the hierarchy of object discriminability observed experimentally (Froudarakis et al., 2020). Given that population response geometry is critical for determining discriminability (Chung et al., 2018; Chung & Abbott, 2021; Li et al., 2024), we investigated whether changes in geometry induced by dropout would influence the hierarchical structure learned by the model.

To explore this, we simulated the experiment of Froudarakis et al. (Froudarakis et al., 2020) using models trained on the MICrONS dataset with varying dropout rates. We then assessed how well each

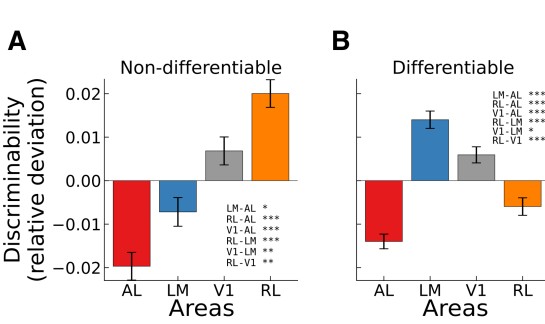

Figure 5: **Hierarchy in discriminability across visual areas in digital twins with different representation geometry** (**A**) Discriminability changes relative to the mean across different areas, computed for the network models with non-differentiable representations. (**B**) Same as in A, but for models with differentiable representations. These models more effectively capture the hierarchical relationships between areas observed experimentally, accurately reflecting the relationships among LM, V1, and RL. However, the model incorrectly places AL at the bottom of the hierarchy. Results in panels A and B were obtained by aggregating simulations from the models shown in Fig. 4, with $\alpha < 1$ for panel A and $\alpha > 1$ for panel B.

model could discriminate between stimuli across different visual areas. Increasing the dropout rate consistently reduced the ability to discriminate stimuli across all areas (Supp. Fig. 5). This finding aligns with prior work, which demonstrates that higher dimensionality, reflected by a lower power-law exponent $\alpha$, facilitates pattern discrimination (Babadi & Sompolinsky, 2014). Additionally, the observed decrease in discriminability may stem from the model's reduced performance in predicting single-neuron responses (Supp. Fig. 5), potentially leading to lower information content in the model's responses to visual stimuli.

Interestingly, dropout also influenced the hierarchy of discriminability between areas (Fig. 5B). In models with non-differentiable representations, object identity was more easily discriminated in RL compared to LM and V1, with AL showing the lowest performance. This is the opposite of what is observed experimentally (Froudarakis et al., 2020) (Fig. 1C). However, when the model exhibited differentiable representations, the hierarchy aligned more closely with experimental data, positioning LM above V1, which was above RL at the bottom (Froudarakis et al., 2020) (Fig. 5B). While these qualitative results are promising, significant discrepancies remain. The model incorrectly places AL at the bottom of the hierarchy, whereas it was found experimentally to be at the top. Additionally, the discriminability values were consistently higher than those observed experimentally, and the differences between visual areas were less pronounced (Supp. Fig. 5). We comment on potential causes for these discrepancies in the discussion section.

## 7 DISCUSSION AND LIMITATIONS

Digital twins have shown promise in modeling single-neuron responses, but their ability to capture population-level response structures remains largely unexplored. In this study, we evaluated their performance in replicating population responses within the mouse visual cortex. Our findings demonstrate that current models struggle to reproduce the differentiability of neural representations and the hierarchy in object discriminability across areas observed experimentally (Stringer et al., 2019a; Froudarakis et al., 2020). We observed this limitation across datasets, model architectures, and loss functions.

Our results show that regularization during training enabled models to replicate the observed differentiability of population responses. This effect likely arises because introducing stochasticity into the relationship between inputs and neural responses fosters a more distributed and robust coding scheme. In our models, stochasticity was introduced through dropout, which randomly deactivates units during training, and data augmentation, which adds variability by shifting and scaling input images. In biological systems, spiking variability and stochastic synaptic transmission effectively introduce an analogous stochastic relationship between inputs and neural representations. Our findings suggest that such biological noise could promote differentiable and robust representations.

The only stimuli that maintained a non-differentiable representation are gratings. This may be because models trained on one domain (e.g., natural images) have limited transferability to different stimulus types, as previous studies have indicated (Sinz et al., 2018). Consistent with this inter-

pretation, we found that the tuning curves predicted by our model resemble those observed experimentally for the oriented stimuli in the MICrONS dataset used for training but not for the gratings used in the experiments by Stringer et al. (2019a) (Supp. Fig. 10). Though recent advancements in foundational models of mouse V1 have improved domain transfer performance (Wang et al., 2023), challenges remain. To properly assess differentiability, it is not enough to predict simple features like orientation preference; the entire tuning curve shape must be considered. This raises important questions about how well models capture the geometric properties of population responses across stimuli.

We demonstrated that models with differentiable representations better captured the hierarchy in object discriminability across areas observed experimentally. Specifically, the discriminability of stimuli increased from RL to V1, and from V1 to LM, consistent with the experimental hierarchy. However, our models incorrectly placed AL at the bottom of the hierarchy and consistently overestimated discriminability and underestimated its variation across areas when compared with precious experimental results (Froudarakis et al., 2020). These discrepancies likely stem from differences between artificial and biological networks, including the absence of spiking noise and reduced trial-to-trial variability in the models. Additionally, differences in training stimuli—such as object movement speed or complexity—may contribute to the observed gaps between model predictions and experimental data.

We found that regularization during training improved the ability to capture population-level responses but reduced single-neuron performance (Supp. Fig. 9), revealing a consistent trade-off across models and regularization schemes. This occurs because state-of-the-art models capture only a finite fraction of the explainable variance for single neurons. Accurately predicting single-neuron responses reflects an approximation of marginal distributions but does not inherently constrain the model to learn the higher-order dependencies that govern population structure and shape the covariance matrix. Furthermore, fitting both single-neuron and population-level responses on one dataset does not ensure generalization, as shown by discrepancies on datasets from Stringer et al. (Stringer et al., 2019a) and Froudarakis et al. (Froudarakis et al., 2020), likely due to overfitting to dataset-specific statistics rather than capturing biologically relevant structures. These results highlight key limitations of current approaches and the need for models that balance this trade-off while generalizing effectively across diverse datasets.

Future studies could benefit from using architectures that differentiate between neurons in distinct visual areas, allowing for a more accurate replication of hierarchical transformations. Indeed, a limitation of our architecture is that the core is shared across all visual areas thereby constraining the model's expressivity. In biological systems, receptive fields expand along the visual hierarchy (Siegle et al., 2021), a property our model does not capture due to its uniform receptive field size. To explore this limitation, we examined more flexible digital twins that allow different areas to rely on distinct internal representations. First, we employed a task-driven modeling approach by finetuning a linear readout from layers of a model trained on object recognition and found that it did not capture the discriminability hierarchy observed experimentally (Supp. Fig. 6). Additionally, we trained a 3D CNN model independently on data from each area to capture area-specific differences. While variations in object discriminability across areas were larger than in our benchmark model, the hierarchy of discriminability was still incorrect (Supp. Fig. 7).

To further improve digital twins, it will be important to move beyond measures of covariance and discrimination and consider more detailed manifold structures that underlie population responses, such as radius and dimensionality (Chung et al., 2018; Froudarakis et al., 2020). These factors, which significantly impact discriminability (Chung et al., 2018; Chung & Abbott, 2021; Li et al., 2024), show clear trends along the visual hierarchy (Froudarakis et al., 2020).

Understanding how complex computations such as object recognition and localization emerge from hierarchical neural transformations is a fundamental challenge. Digital twins provide a promising tool for exploring these questions, as they allow for detailed control over stimuli and response variables. In this work, we have demonstrated that digital twins can capture key aspects of population response geometry and qualitatively reproduce the hierarchy in object discriminability across areas observed experimentally. However, further research is needed to fully validate these models and to explore how joint activity within neuronal populations contributes to computations across different visual areas.

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

# A APPENDIX

CONTENTS

## A.1 NEURAL NETWORK MODEL DESCRIPTION

This section provides a detailed description of the components that make up our benchmark model. We begin by outlining the model architecture, including the key components and how they interact. Specifically, we describe the core, readout, and shifter components, along with the modifications made to the original baseline architecture to optimize performance. Next, we discuss the procedures used to train the model, highlighting the strategies implemented to ensure robustness and accuracy, followed by a description of the metrics we used to evaluate its performance.

### A.1.1 NETWORK ARCHITECTURE

The model architecture is composed of three main components: core, shifter, and readout. The model takes videos and behavioral variables as inputs, which are fed to the "core" component and then to the "readout" component. The readout component linearly combines the representations learned in the last layer of the core (feature weights) and determines the receptive field associated with each neuron (spatial position). The spatial position is further adjusted by the "shifter" which takes as input the pupil center's coordinates. A graphical visualization of the model is provided in Fig. 1A. More details about each element are provided below.

**Inputs.** The input of the neural network model are: (1) Videos and behavior (Tensor V). This is a 5-d tensor with shape (Batch, Channels, Frames, Height, Width) where the first channel represents the gray screen video shown to the mouse and the last 2 channels the behavior (running speed and pupil size). The behavior tensor was expanded to match the Height and Width of the videos and concatenated to the videos tensor in the channel dimension. (2) Pupil center (Tensor P). It is a 3-d tensor with shape (Batch, Frames, 2), with data on pupil center coordinates frame-by-frame.

**Core.** The core, taken from (Höfling et al., 2022), is composed of 4 layers of 3D Convolutional Network (16, 32, 64, 128 channels respectively), factorized in spatial and temporal components. The spatial kernels are $11 \times 11$ in the 1st layer and $5 \times 5$ in all of the subsequent layers. Similarly, the temporal kernels are $11 \times 1$ in the 1st layer and $5 \times 1$ afterwards. After each layer, batch normalization and a final non-linearity (ELU + 1) were applied.

**Readout.** To compute single responses along stimulus presentation, we adopted the readout presented by (Lurz et al., 2020). For each neuron, a learnable spatial position $p_n \in [-1, 1]^2$ is used to apply a bilinear interpolation at specified spatial positions of the core's output. This method extracts a feature vector which is then linearly combined with a learnable feature weight $w_n \in \mathbb{R}^{128}$ and a bias to predict the activity of individual neurons frame-by-frame. An (ELU + 1) non-linearity was applied to final responses to keep them non-negative.

**Shifter.** This component is a multilayer perceptron (MLP) with three layers having five units each that maps pupil center's coordinates into a shift linearly applied to all neurons' positions $p_n$. At the end of each layer, a Tanh nonlinearity is applied to ensure the output remains within the range $\in [-1, 1]^2$.

**Modifications to original architecture.** We slightly modified the original baseline architecture proposed in (Turishcheva et al., 2023) with the following changes: we enforced temporal causality in the temporal convolutions such that representations generated up to a certain time were influenced by previous frames only; we included an additional batch normalization layer between the two convolutions as we found it helped gradient stability and model convergence; lastly, we disabled the Position Network in the readout component, an MLP that maps neurons' anatomical coordinates extracted during recordings to corresponding readout positions $p_n$.

### A.1.2  MODEL TRAINING

**Loss function and training details.** All videos were isotropically downsampled to a resolution of $36 \times 64$ px (H × W) per frame. Functional and behavioral signals were resampled to 30 Hz by linear spline interpolation to match the frequency of visual stimuli. Visual stimuli, behavioral data and pupil center coordinates were normalized by subtracting the mean and dividing by the standard deviation of the corresponding session. Neural responses were standardized to maintain their non-negativity. Behavioral variables (running speed, pupil size) were added as separate channels to the input images (Franke et al., 2022) to incorporate the modulatory effects of behavior on neural activity. Batches consisted of 150 consecutive frames randomly sampled from the corresponding trials (each consisting of 300 frames). We trained our networks with a batch size of 9. For parameter optimization, AdamW was used with a momentum of 0.1, to minimize the negative likelihood Poisson loss:

$$\mathcal{L}_s^{Poisson} = \sum_{t=1}^{n_t} \sum_{i=1}^{n_s} r_{i,t} - o_{i,t}\log(r_{i,t}) \tag{1}$$

between the recorded responses $o$ and predicted responses $r$, where $n_t$ is the number of frames in one batch and $n_m$ is the number of neurons for session $s$. Gradient updates were performed after a full pass across all sessions, a method that has been shown to increase performance (Li et al., 2023). We employed a learning rate scheduler with a linear warm-up over the first 10 epochs. Then, after each epoch, we calculated the correlation to average between predicted and true responses on the validation set and averaged it across all neurons. If the correlation failed to increase for eight consecutive epochs, we stopped the training and restored the model to its state after the best performing epoch. Then, we either decreased the learning rate by a factor of 0.3 or stopped training altogether, if the number of learning-rate decay steps was reached (n=4 decay steps).

**Regularizarion strategies.** When we used data augmentation, we applied random affine transformations (rotation, scaling and translation) to input videos. The transformations were parametrized by a *transformation factor* (here called $\alpha_{\text{affine}}$) determining the amplitude of the transformation applied as follows:

$$\texttt{degrees} = (-\alpha_{\text{affine}} \cdot \texttt{rotation}_{\text{max}},\ \alpha_{\text{affine}} \cdot \texttt{rotation}_{\text{max}})$$

$$\texttt{translate} = (-\alpha_{\text{affine}} \cdot \texttt{translation}_{\text{max}},\ \alpha_{\text{affine}} \cdot \texttt{translation}_{\text{max}})$$

$$\texttt{scale} = (1 - \alpha_{\text{affine}} \cdot \texttt{scale}_{\text{max}},\ 1 + \alpha_{\text{affine}} \cdot \texttt{scale}_{\text{max}})$$

with $\texttt{rotation}_{\text{max}} = 20$, $\texttt{translation}_{\text{max}} = 0.1$ and $\texttt{scale}_{\text{max}} = 0.2$. The parameters $\texttt{translate}$, $\texttt{scale}$ and $\texttt{degrees}$ were used as inputs of the corresponding arguments of the PyTorch class RandomAffine (Paszke et al., 2019).

When we used dropout, we applied it at the end of each layer of the core, so that representations generated were more robust by preventing overfitting and ensuring that the model did not rely too heavily on any specific neurons.

### A.1.3  MODEL EVALUATION

Model evaluation employed three distinct metrics: single trial correlation, correlation to average on repeated trials and normalized correlation coefficient ($CC_{\text{norm}}$). Single trial correlation is calculated as follows:

$$\text{trial corr.}(r, o) = \frac{\sum_{i,j}(r_{i,j} - \bar{r})(o_{i,j} - \bar{o})}{\sqrt{\sum_{i,j}(r_{i,j} - \bar{r})^2 \sum_{i,j}(o_{i,j} - \bar{o})^2}} \tag{2}$$

where $\bar{o}$ and $\bar{r}$ the average recorded and predicted responses across all trials. The correlation to average metric is calculated as follows:

$$\text{avg. corr.}(r,o) = \frac{\sum_i (\bar{r}_i - \bar{r})(o_i - \bar{o})}{\sqrt{\sum_i (\bar{r}_i - \bar{r})^2 \sum_i (o_i - \bar{o})^2}} \tag{3}$$

where $\bar{r}_i = \sum_{j=1}^{J} r_{i,j}$ is the average response across J repeats.

The normalized correlation coefficient $CC_{\text{norm}}$ is another measure evaluated on repeated responses introduced by (Schoppe et al., 2016) to account for the limited number of repetitions and the high variability of neural responses. It is described as follows:

$$CC_{\text{norm}} = \frac{CC_{\text{abs}}}{CC_{\text{max}}}$$

where

$$CC_{\text{abs}} = \frac{\text{Cov}(\bar{r}, \bar{o})}{\sqrt{\text{Var}(\bar{r})\text{Var}(\bar{o})}}$$

and

$$CC_{\text{max}} = \sqrt{\frac{N\text{Var}(\bar{o}) - \overline{\text{Var}(o)}}{(N-1)\text{Var}(\bar{o})}}$$

where $r$ is the predicted response and $o$ is the observed response to $N$ repeated stimuli. $CC_{\text{max}}$ is a normalization factor and it represents the theoretical maximum correlation coefficient between the observed responses $o$ and the model's predicted responses $r$. In this way, it is possible to evaluate neural models in the presence of noise and variability by accounting for the reliability of individual neurons.

## A.2 DETAILS OF SIMULATING THE EXPERIMENT OF STRINGER ET AL. (STRINGER ET AL., 2019A)

**Stimuli composition.** Natural stimuli (natural) were composed by a total of 2,800 stimuli selected from the ImageNet database20, from ethologically relevant categories: 'birds', 'cat', 'flowers', 'hamster', 'holes', 'insects', 'mice', 'mushrooms', 'nests', 'pellets', 'snakes' and 'wildcat'. Spatially whitened (white) versions of the 2,800 natural images, were obtained by adjusting their frequency content using Fourier transforms, scaling them back to the pixel domain, and normalizing the intensity to match the original images' mean and standard deviation. Spatially localized stimuli (small) were obtained by windowing images over the receptive field of the recorded population. Sparse noise stimuli (sparse) consisted of white or black squares on a grey background. The eight- and four-dimensional stimuli (8D, 4D) were constructed using a reduced-rank regression model by projecting the natural images onto a set of $d$ basis functions (eight or four depending on the stimuli constructed). Drifting gratings (ori32) consisted of 32 one-dimensional directions, evenly spaced at 11.25°.

**Experiment replication.** The original experiment consisted in the following procedure: each set of images was presented twice to account for trial-to-trial variability. Images were presented for 0.5 s, alternating with a grey-screen inter-stimulus interval lasting a random time between 0.3 and 1.1 s. Collected responses were averaged across image presentation time to obtain a single response associated to the corresponding image. Ongoing activity dimensions were removed by subtracting the responses' projection onto the top 32 spontaneous dimension. Then, cvPCA was used to estimate the eigenspectrum from the set of responses collected over two presentations. Power law fits were calculated between the 11th and 500th eigenvalues, except for gratings (where the range was 11-30). Since our model was trained on movies, we created videos by repeating images over time (15 frames, corresponding to 0.5 s). Additionally, 50 frames of gray-screen were added before image presentation. Since the model takes as input also behavioral data, these were set to 0 (mean value post normalization) across all stimulus presentation. In this way, we ensured that responses dynamics were determined by visual stimulation only. We then collected the output of the digital twin in response to the image presentation and computed its mean across the 15 frames, obtaining a single response for each image. In line with the original experiment, we focused on responses of neurons coming from V1 at recorded depths 100-350 $\mu$m from cortical surface (corresponding to layer 2/3). Analyses were conducted only on neurons that were well captured by the model, defined as those selected units with a correlation to the average metric greater than 15%. Starting from a dataset

containing responses from all neurons to all stimuli, we calculated the covariance matrix and we applied PCA to obtain the eigenspectrum for each session. From this, we obtained the power law fits using the same methods of the original experiment.

In Fig. 4A and Fig. 4B we report the average power-law exponent over three models trained with different seeds. Panels in Supp. Fig. 3B and in Supp. Fig. 4B show examples for individual models instead.

## A.3 DETAILS OF SIMULATING THE EXPERIMENT OF FROUDORAKIS ET AL. (FROUDARAKIS ET AL., 2020)

To replicate the experiment from Froudarakis et. al. (Froudarakis et al., 2020) with our model, we used Blender (Blender Development Team.) to render four three-dimensional objects (Fig. 1C), replicating those used in the original study. Each object was subjected to a set of continuous, random variations in position (X, Y, Z), scaling, tilt/rotation, and environmental lighting (intensity, position), ensuring smooth and coherent movements consistent across all objects. All renders were done over a uniform gray background. The resulting movies were split into 500 clips of 10 seconds each, rendered at 30 fps, and fed to our benchmark model trained on the MICrONS dataset (Consortium et al., 2021). We extracted the predicted neural responses from layer 2/3 for different visual areas, focusing on neurons well described by the model in the training phase (correlation to average greater than 0.15). We used these extracted responses to construct N-dimensional vectors of neural activity aggregated in 500ms intervals. The vectors were then used to train one-versus-all logistic regression models to classify object's identity for different values of neurons count N. In all cases, we used a 10 fold cross-validation approach and reported the average performances for the 10% of the data that were held out from the training of the decoder. We repeated the procedure 50 times with bootstrap sampling of the neurons at each count step to evaluate decoding accuracy across visual areas. To measure the discriminability we used the mutual information measured in bits by computing

$$\text{MI}(c, \hat{c}) = \sum_i \sum_j p_{ij} \log_2 \left( \frac{p_{ij}}{p_i p_j} \right)$$

where $p_{ij}$ is the probability of observing the true class $i$ and predicted class $j$ and $p_i$ and $p_j$ are the marginal probabilities.

The experiment of Froudarakis et. al. (Froudarakis et al., 2020) characterized the object discriminability across 10 visual areas; in our analyses, we focus on the four (V1, LM, RL, AL) included in the MICrONS dataset (Consortium et al., 2021), which we used to train our model.

## A.4 DETAILS ON VARIATIONS OF THE MODEL DISCUSSED IN SECTION 5.1

**Model trained on neurons with high reliability to repeated videos.** To select the set of neurons whose response were reliable across repeated stimuli, we used the Fraction of Explained Variance (FEV) metric to quantify neurons' reliability. The metric is calculated as follows:

$$\text{FEV}(o) = \frac{\text{Var}_{\text{total}}(o) - \overline{\text{Var}_{\text{noise}}(o)}}{\text{Var}_{\text{total}}(o)}$$

where $\text{Var}_{\text{total}}(o)$ is the variance of responses evaluated across trials, repetitions and frames and $\overline{\text{Var}_{\text{noise}}(o)}$ is the variance evaluated on repeated trials, averaged across trials and frames. For each session, we selected the 50% most reliable neurons.

**Model trained on SENSORIUM.** The SENSORIUM dataset was collected using 5 different mice, all focusing on neurons located in V1. In total, over 38,000 neurons' responses were recorded in response to around 1800 natural scene videos (each lasting 10 s). Behavioral variables (running speed, pupil size and pupil center's coordinates) were recorded along with videos presentation. We repeated the modeling procedure described in Section 3 of the main text on this dataset.

**Transformer-based digital twin.** The only difference of the ViV1T model compared to the benchmark model described in the main text is in the core part. The ViV1T core contains three main components: (1) a tokenizer that concatenates the video and behaviour variables over the channel dimensions and extracts overlapping tubelet patches along the temporal and spatial dimensions, followed by a factorized positional embedding which learns the spatiotemporal location of each patch;

(2) a spatial Transformer which receives the tubelet embeddings and learns the spatial relationship over the patches within each frame; (3) a temporal Transformer receives the spatial embedding and learns a joint spatiotemporal representation of the video and behavioural information. The readout and shifter modules are equal to the model we used.

**Model with loss function on the correlations between neuron.** We developed a new loss function that focused on capturing correlations between neurons rather than accurately predicting single neuron activity. This was defined as:

$$\text{MSE}_{\text{corr}} = \sum_{i=1}^{n_s} \sum_{j=1}^{n_s} (R_{i,j} - O_{i,j})^2 \tag{4}$$

where $R_{i,j}$ is the correlation between predicted responses for neurons $i$ and $j$, while $O_{i,j}$ is the correlation between their observed responses.

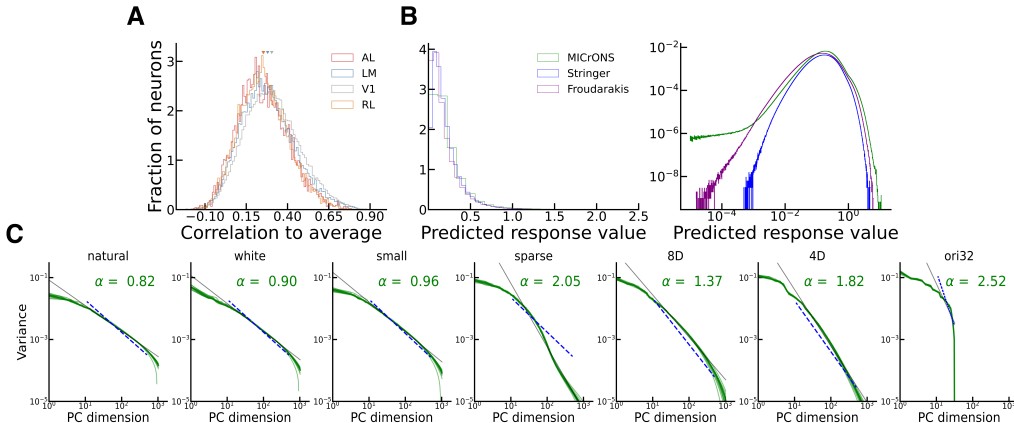

Supplementary Figure 1: **Controls and additional measurements for simulations shown in Fig. 2 (A)** Distribution of correlation to the average measured in the benchmark model for neurons in the four visual areas under investigation. (**B**) Distribution of responses across neurons measured in the benchmark model when simulating the experiments of Stringer et al. (Stringer et al., 2019a) and Froudarakis et al. (Froudarakis et al., 2020). Distributions are shown in linear (left) and logarithmic (right) scale. Despite the visual stimuli being significantly different from those used to train the model, it does not exhibit pathological behavior and generates responses similar to those produced in response to stimuli from the MICrONS dataset. (**C**) Eigenspectrum of the covariance matrix obtained from the digital twin (lighter and darker green lines represent individual sessions and their average, respectively), simulating the experiment by Stringer et al. for all types of stimuli used (columns). In each panel, the dashed line indicates the decay found in the experiment, while the solid gray line represents the best fit of a power law to the model results, with the exponent reported as text.

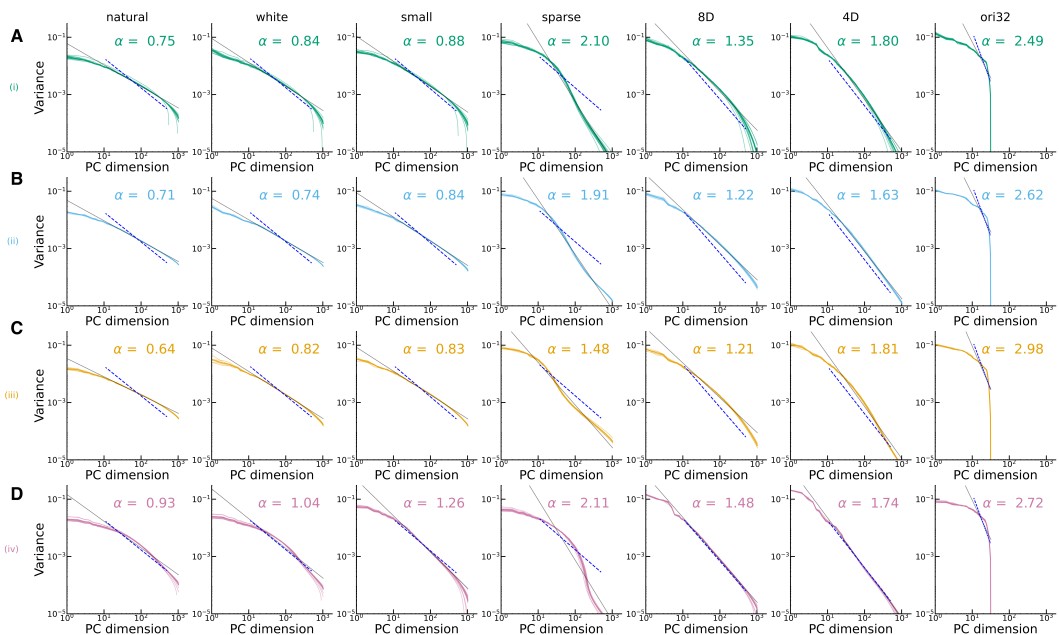

Supplementary Figure 2: **Geometry of V1 population responses across stimuli for the network models shown in Fig. 3.** Eigenspectrum of the covariance matrix obtained when simulating the experiments of Stringer et al. (Stringer et al., 2019a) with: (A) model trained on reliable neurons, (B) model trained on the SENSO-RIUM dataset, (C) ViV1T, and (D) model trained on correlations. Columns correspond to different types of stimuli used (titles). In each row, colored lines represent quantities measured in the digital twins (lighter and darker lines indicate individual sessions and their average, respectively). The dashed lines show the decay observed in the experiment, while the solid gray lines represent the best fit of a power law to the model results, with the exponent reported as text.

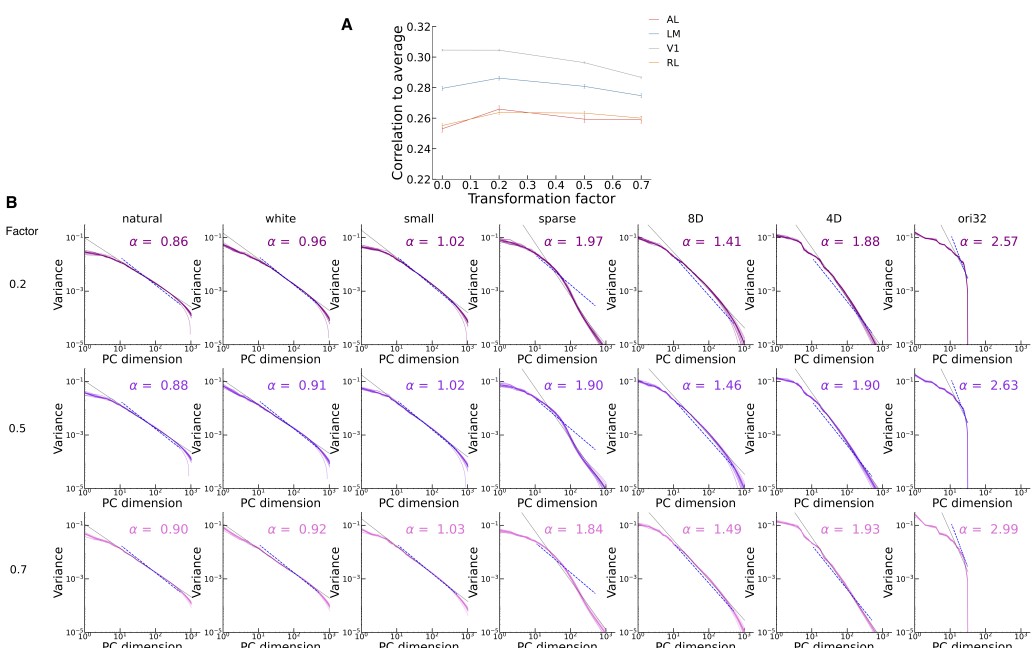

Supplementary Figure 3: **Characterization of responses in simulations underlying the results shown in Fig. 4A** (**A**) Mean correlation to average measured across single neuron responses in models trained with different transformation amplitudes for each analyzed area. (**B**) Eigenspectrum of the covariance matrix obtained from simulating the experiments of Stringer et al.(Stringer et al., 2019a). Different rows correspond to individual models trained with varying transformation amplitudes (text on the left, only models trained with amplitude $> 0$ are shown).

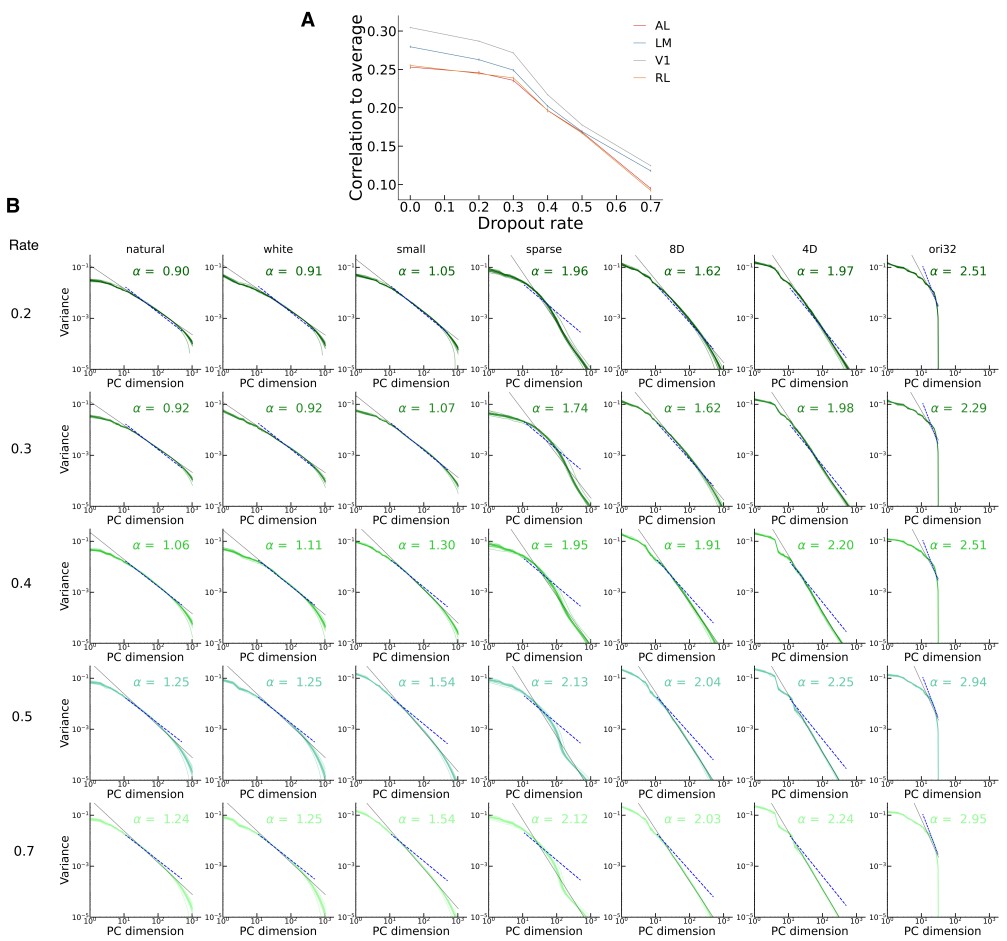

Supplementary Figure 4: **Characterization of responses in simulations underlying the results shown in Fig. 4B,D** (**A**) Mean correlation to average measured across single neuron responses in models trained with different dropout rate for each analyzed area. (**B**) Eigenspectrum of the covariance matrix obtained from simulating the experiments of Stringer et al.(Stringer et al., 2019a). Different rows correspond to individual models trained with varying dropout rate (text on the left, only models trained with dropout rate $> 0$ are shown).

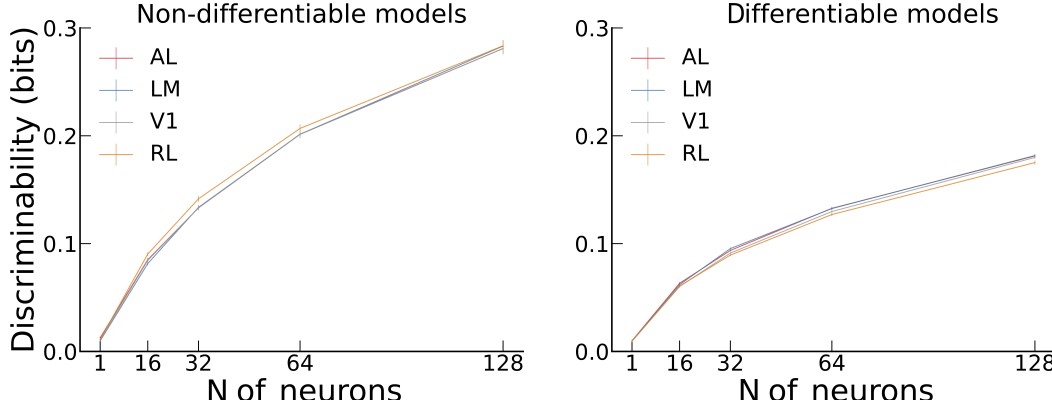

Supplementary Figure 5: **Discriminability in simulations shown in Fig. 5** Discriminability as a function of the number of neurons and cortical areas, computed from the network models shown in Fig.5.

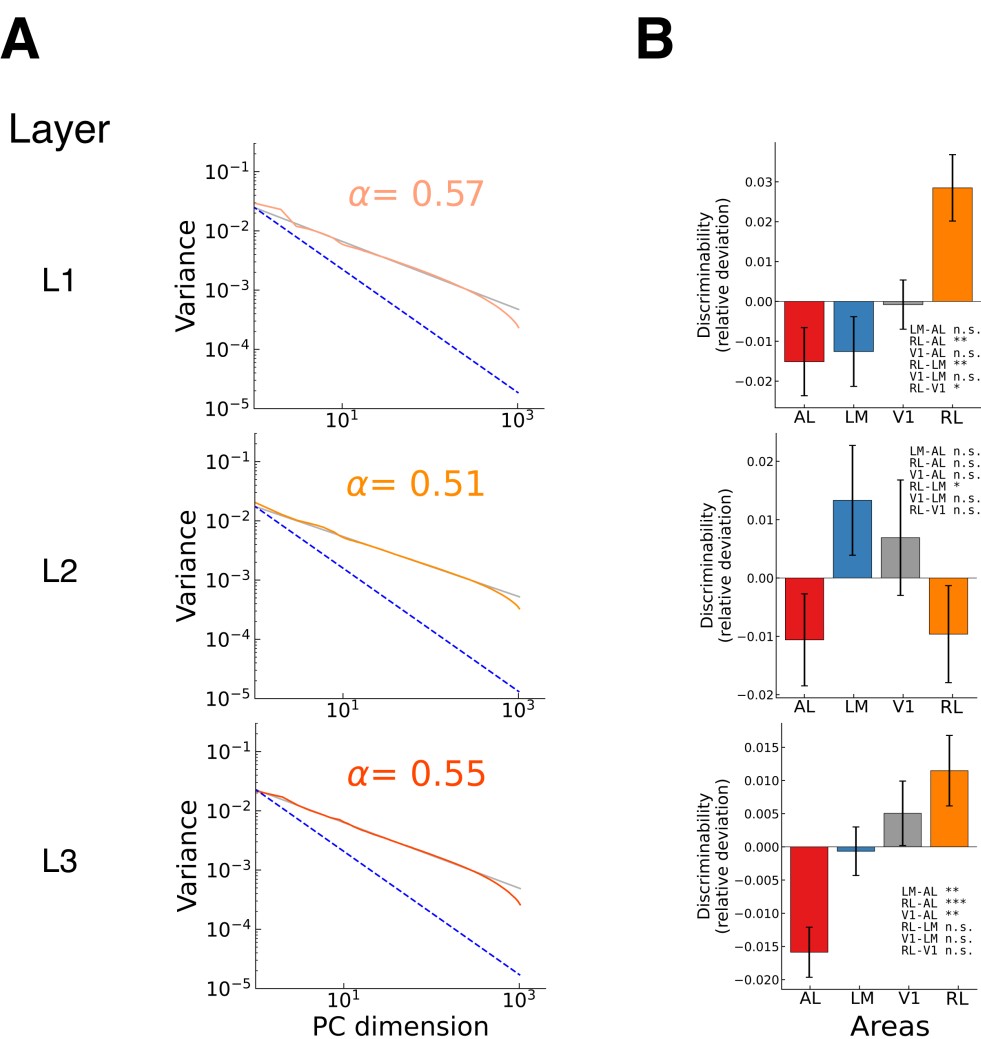

Supplementary Figure 6: **Population geometry in a task-driven model.** We finetuned a ResNet50 model on the MICrONS dataset by training a linear readout after each of the first three layer block while keeping the ResNet50 weights frozen. The same testing procedure used for the data-driven models was applied to the predicted responses. Each row in the figure presents the eigenspectrum of the covariance matrix (left, as in Fig. 2A) and the relative discriminability across areas (right, as in Fig. 2D) based on the linear readouts from each layer. The results were consistent with those from our benchmark model: the learned representation was not differentiable, and the model failed to capture the hierarchy of discriminability observed experimentally.

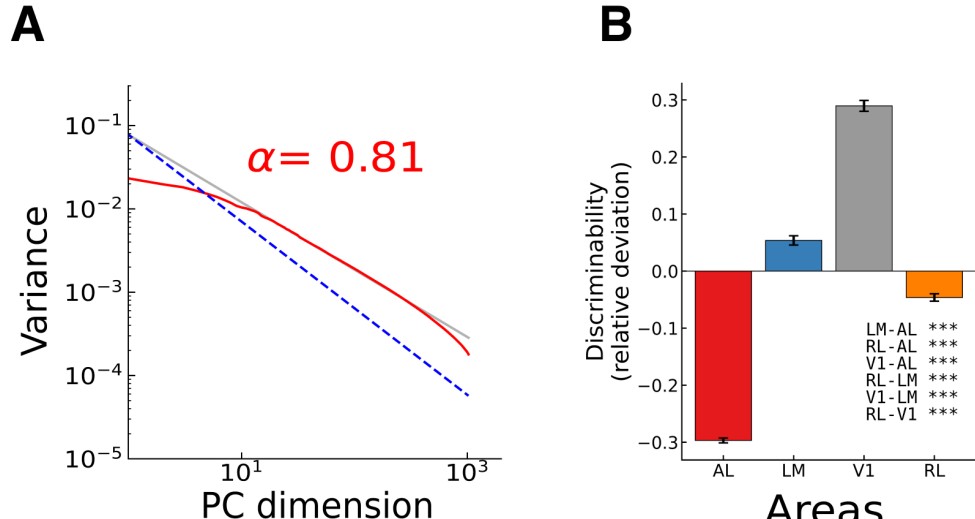

Supplementary Figure 7: **Population geometry in 3D convolution models trained independently for each area.** (**A**) Eigenspectrum of the covariance matrix, as shown in Fig. 2A, but for a 3D CNN model trained exclusively on recordings from V1. The representation learned by the model was not differentiable. (**B**) Relative discriminability across areas, as shown in Fig. 2D, but computed in four 3D CNN models, each trained using data exclusively from one area. Object discriminability across areas varied significantly more than in our benchmark model. While the model correctly positioned V1 above RL in the discriminability hierarchy, it incorrectly placed V1 at the top and AL at the bottom, contradicting experimental observations.

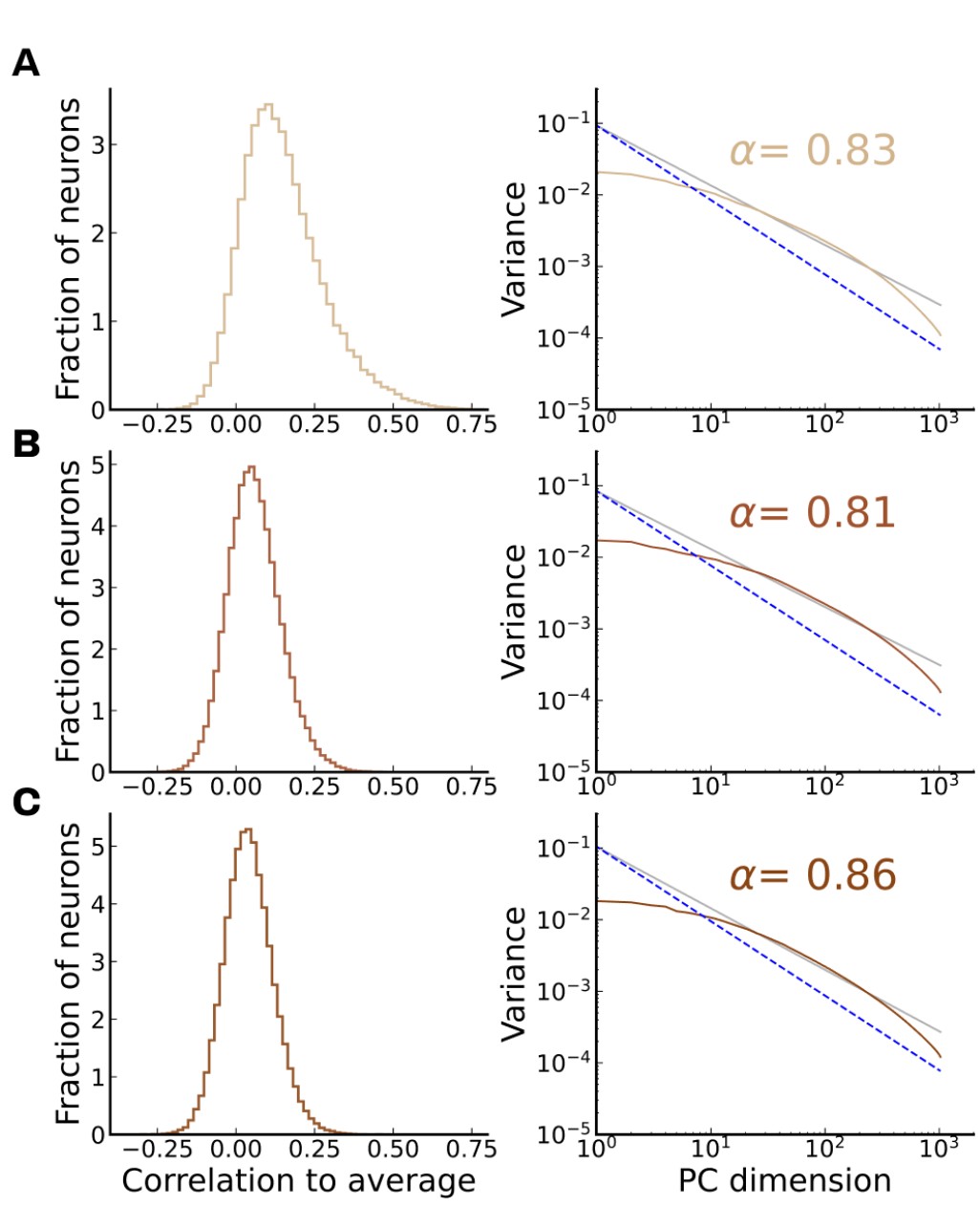

Supplementary Figure 8: **Multi-objective optimization of single-neuron responses and population correlations.** Response properties in network models trained using a multi-objective optimization framework balancing single-neuron responses and population correlations. We trained three models with a combined Poisson single-neuron loss and population correlation loss. Panels show the distribution of correlations and eigenspectrum of the covariance matrix obtained in models with: (**A**) larger weight on the Poisson loss; (**B**) equal contributions from Poisson and population correlation losses; (**C**) larger weight on the population correlation loss. The results showed no evidence of a synergistic effect: gradually increasing the relative weight of the Poisson loss improved the models' ability to capture single-neuron responses but reduced the differentiability of population representations.

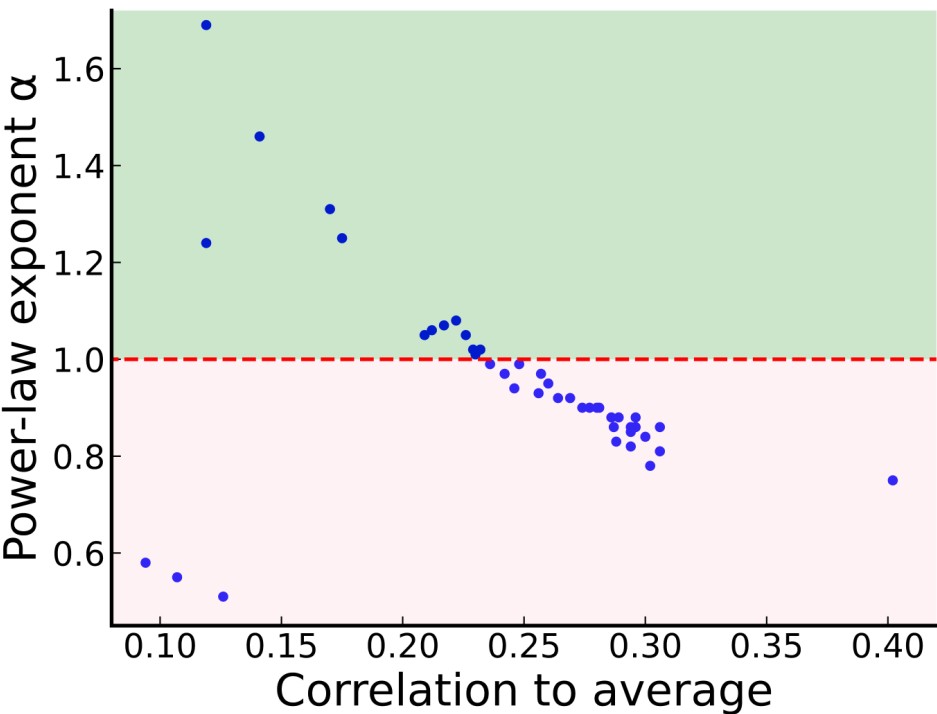

Supplementary Figure 9: **Trade-off between single-neuron responses and population geometry.** This figure shows the power-law exponent of the eigenspectrum of the covariance matrix as a function of the single-neuron correlation to average across all models trained with the Poisson loss. Results reveal a negative correlation between the ability to capture population geometry and the accuracy of single-neuron responses.

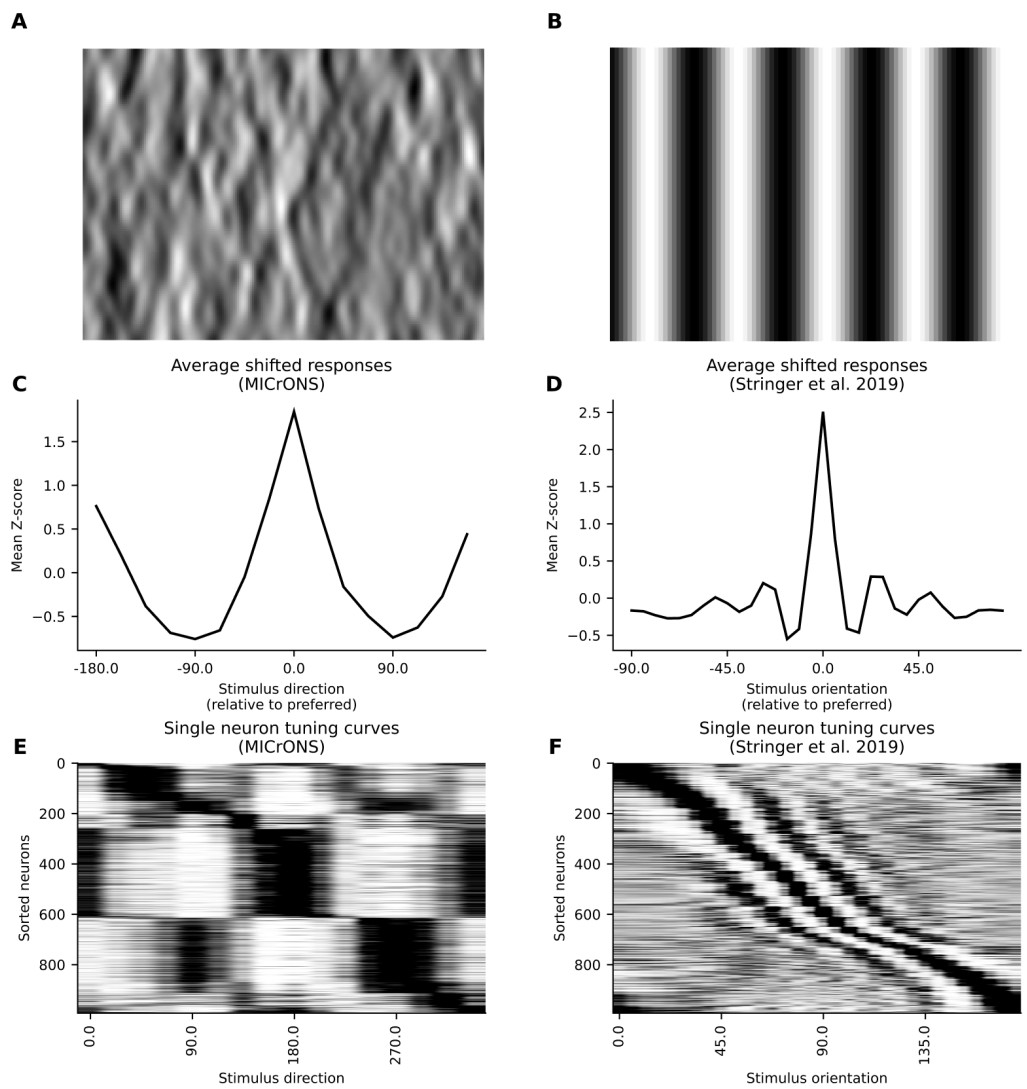

Supplementary Figure 10: **Analysis of model responses to oriented stimuli.** (**A**) Example Monet stimuli from the MICrONS dataset (Consortium et al., 2021), which were used for training the model. (**B**) Example gratings from the experiment by Stringer et al. (Stringer et al., 2019a). These stimuli were not used for model training but were used to test the trained models. (**C**) Average tuning curve of model neurons in response to Monet stimuli. (**D**) Average tuning curve of model neurons in response to gratings. (**E**) Tuning curves of all model neurons sorted by their preferred orientation for Monet stimuli. (**F**) Tuning curves of all model neurons sorted by their preferred orientation for grating stimuli. Responses to Monet stimuli, which were used during training, exhibit tuning curves similar to those commonly observed in real neurons. In contrast, responses to gratings show pathological high-frequency oscillations, likely due to the substantial difference between the grating stimuli and the Monet stimuli used during training.

