# OpenReview forum: "Beyond single neurons: population response geometry in digital twins of mouse visual cortex"
_ICLR.cc/2025/Conference — ICLR 2025 Poster_

### Official Review · Reviewer_A1NT · 2024-11-03

**Soundness:** 3
**Presentation:** 2
**Contribution:** 3
**Rating:** 8
**Confidence:** 2

**Summary:**

This paper argues that some digital twins (i.e., ANN models that predict single neuron activities well) fail to capture population level properties (specifically having to do with the representational geometry) of neural responses that have been previously observed in the mouse visual cortex. The primary issue is the fact that the population representations in the digital twins have a geometry that is non-differentiable, which suggests that small changes in inputs can lead to very large changes in the network's representation. The paper then shows that dropout regularization can go some way towards mitigating these kinds of discrepancies between the ANN's responses and those of the actual mouse brain.

**Strengths:**

Understanding how highly predictive ANN models of the brain can fail to capture certain properties of neural population responses that are functionally important is an important issue to address. This paper exposes some shortcomings of existing models, and gives a fairly straightforward way of mitigating some of those shortcomings, using dropout in the network. This is useful because it may lead to models that do a better job of accurately simulating brain responses.

**Weaknesses:**

The paper could be clearer in a lot of places, especially for readers who are not familiar with the model being used, as well as the background literature which the paper heavily draws upon.
- Although the model is described in the Appendix, it wasn't very clear (at least, not to me). There are a lot of different parts, and it was hard to see how they fit together. It might help to have a diagram illustrating the architecture and how everything fits together.

Although a lot of the background theory relating to the differentiability of neural representational geometry is presented in Stringer 2019, it would help readers if this paper were a bit more self contained, e.g. some intuitive explanation of why the cutoff for differentiability is 1 + 2/d would have been helpful to the uninitiated reader.

Training the model only on correlations between neurons resulted in a loss of single neuron predictivity, presumably because single neuron predictivity is no longer part of the optimization objective. I would have thought a natural thing to try would be to train the model on a combined objective that optimizes for both objectives at once, and see if that changes the results in any way.

The claim that the digital twins fail to capture the hierarchical relationships between areas seemed like an overstatement, since what is really shown here is that one aspect of hierarchical relationships was not captured, not that it doesn't capture any relevant aspects of the visual hierarchy.

**Questions:**

I was a bit confused at a high level as to how it is even possible for a digital twin to do a good job at predicting single neuron responses and yet still not capture population structure. Doesn't the population structure depend on the single neuron responses? How can you capture each individual neuron's activity with a high degree of accuracy and not get the population structure right?

Questions about the model:
- I was very surprised to read in the paper that the model does not have increasing receptive field sizes. Isn't that just a standard feature of a CNN?
- What is the intuition behind the formula for CC_max (where does this formula come from)?

---

> ### Author Response · Authors · 2024-11-23
> **Response to Reviewer A1NT, part 1/2**
>
> Thank you for your careful evaluation and constructive comments.
>
> ### **Improving clarity of the paper (first weakness):**
>
> We agree with the reviewer that the paper would benefit from an expanded discussion of the previous literature. To address this, we have made the following modifications:
>
> 1) **Model description.** We have restructured and expanded the description of the model in Appendix A1. To improve readability, we have restructured the section, dividing it into architecture, training procedure and evaluation. We have also added a reference to Figure 1A, which illustrates the architecture and how the various components fit together, that was missing in the previous version of the manuscript.
> 2) **Expand explanation of experimental results.** We expanded the description of the experiments by Stringer et al.(2019) and Froudarakis et al.(2021) in the introduction of the manuscript. Regarding the intuition behind the cutoff for differentiability at $1 + 2/d$, it is challenging to provide a simple explanation, as the specific value of $1 + 2/d$ is derived through functional analysis and is not directly intuitive from the original papers.
> To facilitate understanding without requiring a detailed reading of the original work, we have added further clarifications.  Stringer et al. noted that a slow decay requires too many neurons for stimulus reconstruction and makes responses overly sensitive to small changes, hindering generalization. Additionally, through functional analysis, they showed that $\alpha$ reflects the geometry of neural representations: dimensionality increases with $\alpha$, and differentiability depends on key thresholds: $\alpha < 1$ is discontinuous, $1 \leq \alpha < 1 + 2/d$ is continuous but not differentiable, and $\alpha > 1 + 2/d$ is differentiable. These results stem from the relationship between the eigenspectrum and neural response vector similarity. A slow decay causes instability by making the covariance matrix trace diverge.
>
> ### **Train the model on a combined objective (second weakness):**
>
> We followed the suggestion of the reviewer and developed a multi-objective optimization that aims to capture both single-neuron responses and the correlation of the neural population. To test this, we trained three models with a combined Poisson single-neuron loss and population correlation loss, varying the relative weights of the two components (equal weighting or biasing toward one objective). However, we found no evidence of a synergistic effect. We found that gradually increasing the relative weight of the Poisson loss led to models better capturing single-neuron responses but at the expense of the differentiability of population representations.  These results are now discussed in Section 5.1, and we have added a supplementary figure to illustrate our findings (Supp. Fig. 8).
>
> ### **Adjust claims on limitations the model digital twins fail to capture the visual hierarchy (thirds weakness):**
>
> We agree with the reviewer that hierarchy is a complex phenomenon and has multiple perspectives. In our work, we highlight only one aspect that is the discriminability of objects. Following the reviewer’s suggestion, we have updated the manuscript clarifying what aspect of the hierarchical relationship our models do not capture in the text. This was done in: caption of Fig. 2, section 4, caption of Fig. 5, discussion, appendix A.3
>
> ### **Relationship between accuracy at single neuron and population level (first question)**
>
> We appreciate the reviewer highlighting this important point and aim to clarify it further. Specifically, we trained our models on the MICrONS dataset and tested them on two additional datasets: Stringer et al. and Froudarakis et al. Perfectly predicting neural responses during training would likely result in alignment between single-neuron predictions and population-level properties. However, state-of-the-art models typically capture only ~50% of the explainable variance, leaving room for discrepancies.
>
> Importantly, accurately predicting single-neuron responses reflects an approximation of each neuron's marginal distribution. However, population structure arises from the joint distribution of neuron activities, which governs the covariance matrix. Capturing single-neuron responses does not inherently constrain the model to learn these higher-order dependencies. Additionally, the ability to predict single-neuron responses and covariance on the training dataset does not guarantee generalization to different datasets, as models may overfit to dataset-specific statistics rather than learning biologically relevant population structures. This limitation likely explains the observed trade-off between predicting single-neuron responses and accurately capturing population-level correlations.
>
> We have emphasized this point in the discussion of the manuscript.

---

> ### Author Response · Authors · 2024-11-23
> **Response to Reviewer A1NT, part 2/2**
>
> ### **Receptive field size across areas the model (second question)**
>
> We thank the reviewer for their observation. We used a state-of-the-art model from the Sensorium competition, which consists of 3D convolutional layers with fixed spatial and temporal kernel sizes (11×11 and 5×5 spatial kernels, 11×1 and 5×1 temporal kernels). While the receptive field size of units in these models increases across layers, neurons across all areas are modeled exclusively using units in the final layers of the CNN, as is standard in the field. Consequently, all neurons in the dataset are modeled by units in the CNN with the same receptive field size. When this architecture was introduced to describe visual responses, its primary aim was to capture local spatiotemporal patterns in the video input, which are thought to be highly relevant for predicting neural responses in early sensory areas. The fixed kernel sizes emphasize preserving high-resolution local features rather than aggregating global spatial or temporal context. However, we recognize that this lack of increasing receptive field sizes may limit the model's ability to capture more global or long-range dependencies, which could be critical for certain neurons or stimuli. Our findings, specifically the failure of our models to quantitatively account for the hierarchy in object discriminability observed experimentally across areas, suggest that this limitation is indeed significant and that alternative architectures should be explored. In the revised manuscript, we have included additional analysis using task-driven models, allowing different visual areas to be associated with distinct layers of the CNN and thus different receptive field sizes. This approach represents a first step toward addressing this limitation, but further work will be necessary.
>
> ### **Intuition behind the formula for CC_max (third question)**
>
> The $CC_{max}$ is a component of the $CC_{norm}$ metric which is a standard evaluation measure, widely used in the neuroscience community and competitions like SENSORIUM. It was firstly introduced by [1], where $CC_{norm}$ was used to evaluate neural models in the presence of noise and variability. Specifically, it introduces a normalization step that accounts for the reliability of individual neurons which is crucial to consider their trial-to-trial variability. Conceptually, it answers the question: how well could a perfect model correlate with the data, given the intrinsic noise of the neuron?  In more detail, $CC_{max}$ represents the theoretical maximum correlation coefficient between the observed firing rate y and the model's predicted firing rate $\hat{y}$​. It is derived as the correlation coefficient between the observed firing rate $y$ (across N trials) and the true underlying firing rate function (the "ground truth" neuronal response) that would be observed in a perfectly stationary system with an infinite number of trials ($N \rightarrow \infty $). We have updated the text in the appendix to clarify this point (See A.1: Model evaluation).
>
> [1]: Schoppe O. et al. "Measuring the performance of neural models." Frontiers in computational neuroscience (2016).

---

> > ### Comment · Reviewer_A1NT · 2024-11-26
> > **Most concerns now addressed**
> >
> > Thanks to the authors for addressing my concerns. Most of my concerns have been addressed now. However I still do not follow the authors' explanation of why there can be a tradeoff between single neuron predictivity and capturing population level responses. I agree there can be room for discrepancies at only 50% explained variance, but as the authors point out, getting close to 100% explained variance likely *would* result in capturing population level responses as well, so it's not clear still why there should be this systematic tradeoff. I will revise my rating to "Accept".

---

> ### Author Response · Authors · 2024-11-28
>
> Dear Reviewer A1NT, We sincerely thank you for taking the time to review our responses and for your thoughtful reconsideration. We are grateful for your constructive feedback and are glad that our clarifications resolved most of your concerns.
>
> Regarding the tradeoff between single-neuron predictivity and capturing population-level responses, in our setting we believe this arises because models with regularization introduce noise-corrupted signals during training. This effectively reduces the information available to the digital twin to replicate individual activity (resulting in greater robustness of the responses at population-level). The larger the regularization, the greater the noise introduced.
> However, we acknowledge that this tradeoff does not need to hold systematically, and models close to capturing 100% of explainable variance might lead to a more aligned population representation. This point requires a deeper understanding and merits further investigation in future work, for example by exploring more innovative architectural changes that better capture neurons' relationships.
>
> Thank you once again for your valuable input and for revising your rating.

---

### Official Review · Reviewer_sa6o · 2024-11-03

**Soundness:** 2
**Presentation:** 3
**Contribution:** 3
**Rating:** 5
**Confidence:** 4

**Summary:**

The study examines how well digital twins, computational models trained on neural data, capture both single-neuron and population-level responses in the mouse visual cortex. While these models predict single-neuron activity accurately, they fail to replicate the complex geometry and hierarchical organization of population responses seen in biological data. Factors like dataset composition and network architecture didn't resolve these issues. Introducing regularization techniques like dropout improved the models' ability to create more accurate population representations but reduced single-neuron prediction accuracy. The study highlights the need to refine digital twins to better understand neural computations at a population level.

**Strengths:**

The paper tackles an important gap in computational neuroscience by focusing on the geometry of population-level neural responses in digital twins, not just single-neuron predictions. The study meticulously replicates key experimental findings and explores how factors like dataset variation, architectural changes, and training objectives impact model performance. Regularization techniques, especially dropout, are shown to improve the models' alignment with biological data. The paper is well-explained and accessible, providing valuable insights for both neuroscience and machine learning. By identifying the limits of current models, the research sets the stage for developing more accurate and biologically realistic models, emphasizing the need for principles like robustness and generalization in neural computation replication.

**Weaknesses:**

Despite improvements with regularization, the models still show quantitative differences from experimental data, especially in discriminability measures and visual area hierarchy, warranting a deeper analysis to uncover the causes. There's a trade-off where enhancing population-level representation reduces single-neuron prediction accuracy, raising practical concerns. The paper's focus on a shared-core architecture limits insights; exploring more varied architectures could better reflect the visual cortex's hierarchical nature. Additionally, the study relies on computational simulations, and direct physiological validation of these models would further substantiate the findings.

**Questions:**

Have the authors considered multi-objective optimization to balance single-neuron accuracy and population-level representation? Could adding biological features like spiking noise, synaptic dynamics, or neural variability improve the models' population properties? Have they explored area-specific modules or varied receptive field sizes to better capture hierarchical processing? Can they comment on the models' generalizability to different visual stimuli, given the issues with non-differentiable representations for gratings? Would using spiking neural network models help capture population response geometry and address trial-to-trial variability?

---

> ### Author Response · Authors · 2024-11-23
> **Response to Reviewer sa6o, part 1/3**
>
> We appreciate the reviewer’s thoughtful feedback and their positive remarks on the clarity of our exposition. Below, we address each of the points raised:
>
> ### **Analysis of the causes of quantitative differences in discriminability (first and second weaknesses, and third question):**
>
> We agree with the reviewer that the quantitative discrepancies in object discriminability between the data and the model warrant deeper analysis. To address this, we took the following steps:
> 1) **Robustness of results.** Since the changes in discriminability across areas were relatively small, we evaluated the robustness of our findings by repeating simulations with multiple seeds. In the updated manuscript (Figure 5), we present results from 13 models with power-law exponent $\alpha>1$. This extends the previous analysis, which included only 3 models, and addresses potential discrepancies due to limited sampling. The simulations confirm that the model correctly captures the hierarchical relation between LM, V1, RL. They also reveal that the model incorrectly places AL at the bottom of the hierarchy, rather than at the top. Moreover, small differences in discriminability remain robust across realizations, suggesting these issues are fundamental to the model.
> 2) **Task-driven models.** Building on the reviewer’s suggestion to explore different architectures, we trained task-driven models with linear readouts added after each of the first three layer blocks. This approach allowed neurons in different areas to access distinct representations of the stimuli. We incorporated area-specific weights for each module and analyzed the resulting hierarchy of discriminability.  We found results similar to our benchmark model, i.e.  the model did not capture the magnitude and hierarchy of  discriminability observed experimentally across areas. These findings are now shown in Supp. Fig. 6. We updated the manuscript and described the task-driven results in section 5.1 and in the discussion.
> 3) **Area-specific modules.** As suggested by the reviewer, we explored the impact of area-specific modules on discriminability. We trained four copies of our benchmark architecture independently on data from different areas and then tested their hierarchy of discriminability. Variations in object discriminability across areas were markedly larger than those in the benchmark model. While the model correctly ranked V1 above RL in the discriminability hierarchy, it incorrectly placed V1 at the top and AL at the bottom, deviating from experimental observations. These results indicate that training models on data from a single area can capture certain hierarchical relationships but falls short of fully reproducing experimental discriminability patterns. They are now shown in Supp. Fig. 7; we will describe them  in section 5.1 and in the discussion.
> 4) **Characterization of representational geometry.** To further investigate these discrepancies, we analyzed the geometry of object representations. When an object undergoes identity-preserving transformations, it generates a “manifold” of responses in neural space. We characterized these manifolds by measuring their typical size and radius, similar to analyses conducted on experimental data by [1,2]. Experimental results have shown that the radius and dimensionality of these manifolds decrease along the visual hierarchy. Our analysis revealed that the radius of the manifolds in our simulations was comparable in magnitude to experimental observations. However, the manifold dimensionality in our simulations was one order of magnitude smaller than in the experiments (see Fig. 4D of [1]). Moreover, the differences across areas in our simulations were significantly smaller than those observed experimentally. The results of this analysis are summarized in the table below.
>
> | Metric    | AL                   | LM                   | V1                   | RL                   |
> |---------------|--------------------------|--------------------------|--------------------------|--------------------------|
> | Radius    | $1.4041 \pm 0.0341$      | $1.4231 \pm 0.0337$      | $1.4490 \pm 0.0375$      | $1.4121 \pm 0.0323$      |
> | Dimension | $14.9679 \pm 0.6944$     | $15.1874 \pm 0.8668$     | $15.3639 \pm 0.7657$     | $15.1756 \pm 0.9001$     |
>
> While these results do not fully resolve the discrepancies between the experimental data and the model, they provide a more detailed characterization of the underlying factors. Our analyses also rule out certain alternative explanations and offer a foundation for future investigations aimed at addressing these issues.
>
> [1]  Froudarakis E. et al. Object manifold geometry across the mouse cortical visual hierarchy BioRxiv (2020).
>
> [2]  Chung SY. et al. Classification and geometry of general perceptual manifolds. Physical Review X (2019).

---

> ### Author Response · Authors · 2024-11-23
> **Response to Reviewer sa6o, part 2/3**
>
> ### **Tradeoff in capturing single neuron and population responses (second weakness)**
>
> Analyzing all trained models, we confirmed, as noted by the reviewer, a clear negative correlation between capturing single-neuron responses and population geometry. To illustrate this tradeoff, we added a new figure (Supp. Fig. 9) and referenced it in the discussion. This finding underscores two key implications. First, while accurate single-neuron predictions reflect an approximation of marginal distributions, population geometry depends on joint distributions governing covariance structure. Capturing single-neuron responses alone does not inherently constrain models to learn these higher-order dependencies, which likely explains the observed tradeoff. Additionally, models trained on a single dataset may overfit to dataset-specific statistics, limiting their ability to generalize biologically relevant population structures across datasets, as we observed when testing on Stringer et al. and Froudarakis et al.
>
> However, we are uncertain about the specific meaning of "practical concerns" mentioned in the reviewer’s comment. Based on our interpretation (please let us know if this is incorrect), this finding has two key implications. First, the choice of architecture and training procedure should align with the study's focus. For example, models optimized for single-neuron responses are better suited for tasks like identifying the most exciting stimulus for specific neurons [1], while our findings suggest that implementing regularization strategies is critical for capturing population-level computations. In the long term, achieving accurate population geometry without compromising single-neuron performance remains a crucial goal. Our results indicate that balancing these competing demands will require revisiting current architectures and training procedures.
>
> [1]: Walker E. Y. et al. "Inception loops discover what excites neurons most using deep predictive models." Nature neuroscience (2019)
>
> ### **Lack of physiological validation (fourth weakness)**
>
> Although our analysis is mainly computational, we ground our results on biological validation. We agree with the reviewer that there is a lack of large-scale recording to test neural population properties and we hope our work will raise awareness about the importance of measuring and modeling population activity.
>
> ### **Multi-objective optimization (first question)**
>
> In the previous manuscript, we showed that training models solely on population correlation loss improved representation differentiability but couldn’t predict single-neuron responses as it was not part of the loss. We followed the suggestion of the reviewer and developed a multi-objective optimization that can capture both single-neuron responses and the correlation of the neural population. To test this, we trained three models with a combined Poisson single-neuron loss and population correlation loss, varying the relative weights of the two components (equal weighting or biasing toward one objective). However, we found no evidence of a synergistic effect since  gradually increasing the relative weight of the Poisson loss led to models better capturing single-neuron responses but at the expense of the differentiability of population representations. These results are now discussed in Section 5.1, and we have added a supplementary figure to illustrate our findings (Supp. Fig. 8).

---

> ### Author Response · Authors · 2024-11-23
> **Response to Reviewer sa6o, part 3/3**
>
> ### **Effects of including biological features in the models (second and fifth questions)**
>
> These questions raised an important point that we had only partially considered previously. With respect to trial-to-trial variability, experiments have shown approximately 20% of the variance can be explained by behavioral variables [1]. Our models, consistent with previous work, partially account for this variability by including running speed and pupil size as inputs that modulate neural responses on a single-trial basis. Our analysis suggests that such behavioral modulation does not influence the geometric properties of the learned representations, as these remain invariant when the behavioral inputs are removed (Fig. 3). However, it remains to be tested whether incorporating additional behavioral inputs, such as facial movements (as in Stringer et al., 2019), would alter these conclusions.  With respect to spiking noise and synaptic dynamics, we showed that specific regularization schemes, such as dropout in the core or data augmentation at the stimulus level, encourage the network to learn representations more closely aligned with biological data. This suggests that introducing noise at various stages in the network could promote differentiable representations. In biological systems, sources of noise are abundant. For example, spiking variability and stochastic synaptic transmission could all have similar effects. These ideas remain speculative and  future work should explore how the strength of such noise impacts performance. We have added a comment on this in the discussion section of the paper.
>
> [1]: Stringe C. et al. "Spontaneous behaviors drive multidimensional, brainwide activity." Science (2019)
>
> ### **Models' generalizability to different visual stimuli (fourth question)**
>
> Digital twins of the visual system often struggle to generalize to out-of-domain stimuli. Prior work (e.g., [1]) shows that models trained only on natural movies perform worse on grating responses compared to those trained on both natural movies and gratings. Motivated by this, we trained models on both natural movies and structured noise stimuli with orientation (Monet, from the MICrONS dataset). However, as the reviewer noted, models that learn a differentiable representation for natural images still fail for standard gratings. We attribute this to differences between the stimuli used in training and those used in Stringer et al., which we employed to assess differentiability. While Stringer et al. presented standard gratings, MICrONS used structured noise. Analyzing the model’s responses, we found them to show reasonable tuning curves as a function of the stimulus orientation (MICrONS) but produce high-frequency fluctuations for gratings (Stringer). These fluctuations differ significantly from experimental responses observed in Stringer et al. (not used in training), leading to higher-dimensional representations and non-differentiable features. Including standard gratings in training could resolve this limitation. We have added these findings to the discussion and Supp. Fig. 10.
>
> [1]: Sinz F. et al. "Stimulus domain transfer in recurrent models for large scale cortical population prediction on video." Advances in neural information processing systems (2018)

---

> > ### Author Response · Authors · 2024-11-28
> >
> > Dear Reviewer sa6o,
> >
> > We sincerely appreciate your detailed review and the valuable feedback you provided on our submission.
> >
> > We would like to kindly request that you review our responses to confirm that we have adequately addressed your comments and concerns. If additional experiments or clarifications are needed, we would be happy to incorporate them—please let us know promptly to ensure we have enough time to make the necessary adjustments.
> >
> > If you have any further questions or suggestions, please do not hesitate to contact us. Should you find our responses satisfactory, we hope you might consider improving your evaluation and recommending the paper for acceptance.
> >
> > Best regards,
> > Authors of Submission11170

---

> > > ### Author Response · Authors · 2024-12-02
> > >
> > > Dear Reviewer sa6o,
> > >
> > > Thank you for your thoughtful feedback. We have carefully addressed your comments and kindly invite you to review our responses. Please let us know if further clarifications or adjustments are needed.
> > >
> > > Best regards,
> > > Authors of Submission11170

---

### Official Review · Reviewer_YdfR · 2024-11-08

**Soundness:** 3
**Presentation:** 3
**Contribution:** 3
**Rating:** 6
**Confidence:** 4

**Summary:**

The authors demonstrate that a reasonably successful model of visual cortical neurons does not capture population metrics.  They show that measures of the population geometry and visual hierarchy fail on this model.  The eigenspectrum of the covariance matrix does not show scaling consistent with differentiability and the relative discriminability across areas does not match what is observed empirically.  The authors investigated various aspects of the model and of training to see what effect they had on these measurements.  These include selecting for reliability, choices of dataset, dataset augmentation, model architecture, and using a covariance-based loss function.  All of those failed to show different results for the population metrics.  The approach that did produce differentiable representations was adding dropout during training, which showed an exponent consistent with differentiability for sufficiently high dropout rates.  They claim that the models that show differentiability also show the appropriate hierarchical relationships for discriminability.

**Strengths:**

This paper is reasonably clear and well-written.  The question is a very important one as population level characteristics of neural activity are undoubtedly important for neural computation and models need to be able to capture it.  It is extremely important to point out that our current state-of-the-art models are not meeting this criteria and likewise important to address what aspects of training will correct this.  The authors tested a broad set of training procedures.

**Weaknesses:**

The authors did an excellent job of testing an array of procedures, but the results are still unfortunately limited by the number of models and data sets available for testing.  This is not terribly actionable, sadly, in a practical sense.

The authors wish to make a general claim about regularization, but only test dropout.

**Questions:**

Does this result hold for task-trained models as well?

Can you test other forms of regularization?  Do they have the same effect, or is dropout special?

---

> ### Author Response · Authors · 2024-11-23
> **Response to Reviewer YdfR, part 1/2**
>
> We thank the reviewer for their insightful feedback and for appreciating the clarity of our exposition. We would like to address the specific points raised.
>
> ### **Limited number of model and datasets (first weakness and first question):**
>
> Our manuscript aims to highlight a critical gap in the literature: the predominant focus on training models to predict single-neuron responses, often overlooking the significance of capturing responses at the population level. To support our claim, we specifically used state-of-the-art models with high single-neuron prediction accuracy and tested them on two primary datasets:
> - **Data from Stringer et al.** – We used this dataset to analyze the covariance structure of neural responses and its variation depending on stimulus type and dimensionality.
> - **Data from Froudarakis et al.** – This dataset allowed us to assess the hierarchy of discriminability across visual areas.
>
> In response to the reviewer’s concern, we have extended the set of models in our analysis to include:
> - **Task-driven models.** We finetuned a ResNet50 model on the microns dataset. Specifically, we trained a linear readout after each of the first three layer blocks keeping the weights of the ResNet50 model frozen. We applied our testing procedure on the predicted responses following the same approach we used for the data-driven models. We found results similar to our benchmark model, i.e. the learned representation was not differentiable and the model did not capture the hierarchy of discriminability observed experimentally. These findings are now shown in Supp. Fig. 6. We updated the manuscript and described the task-driven results in section 5.1 and in the discussion.
>
> - **Models trained independently for each area.** Starting with our benchmark architecture--a 3D CNN--we repeated the training procedure four times, each using data exclusively from one area. This approach was designed to enable the model to better capture differences across areas. The representation learned by the model trained on V1 data alone was not differentiable ($\alpha=0.81$). Notably, variations in object discriminability across areas were significantly larger than those observed in our benchmark model. Regarding the discriminability hierarchy, the model correctly positioned V1 above RL; however, it placed V1 at the top of the hierarchy and AL at the bottom, which contradicts experimental observations. These findings suggest that training models on single-area data can capture some hierarchical relationships but fails to account for the differentiability of V1 representation and to fully replicate experimental discriminability patterns. They are now shown in Supp. Fig. 7; we described them  in section 5.1 and in the discussion.
>
> - **Multi-objective optimization.** In the previous version of the manuscript, we demonstrated that training models solely on population correlation loss improved representation differentiability but failed to predict single-neuron responses. To extend these findings, we developed a multi-objective optimization framework that combined a Poisson single-neuron loss and a population correlation loss. We trained three models with varying relative weights for these objectives. Our results revealed no evidence of synergy: gradually increasing the relative weight of the Poisson loss led to models better capturing single-neuron responses but at the expense of the differentiability of population representations. These results are shown in Supp. Fig. 8 and discussed in Section 5.1.
>
> The first two models introduce greater flexibility compared to those in the previous manuscript, allowing different areas to rely on distinct internal representations. While this flexibility accounts for inter-area differences, it does not substantially improve the ability to capture the hierarchical organization across regions. The multi-objective optimization, aimed primarily at achieving a differentiable model, ultimately fell short of this goal.  With the addition of these new models, we have significantly expanded the range of approaches investigated in this study.
>
> We agree with the reviewer that there is a lack of large-scale recording to test neural population properties and we hope our work will raise awareness about the importance of measuring and modeling population activity, informing both computational and experimental research in neuroscience.

---

> ### Author Response · Authors · 2024-11-23
> **Response to Reviewer YdfR, part 2/2**
>
> ### **Generality of the statement about regularization (second weakness and second question):**
>
> In our study, we demonstrated that a specific form of regularization--dropout--improved the alignment of population-level properties between experimental data and network models. In the updated version of our manuscript, we extend these findings by showing that similar results can also emerge through data augmentation. Although this approach was present in the previous version, we have found that a larger increase in the magnitude of transformations makes the representation learned by the model differentiable. These results suggest that our observation—that regularization improves alignment with experimental data—is not specific to dropout but instead emerges more generally when strategies to enhance the robustness of representations are employed. This finding is reminiscent of the robustness-to-noise principle often invoked in normative models of biological circuits. The new results on data augmentation are shown in Figure 4 and described  in Section 5.2 and in the Discussion.

---

> > ### Author Response · Authors · 2024-11-28
> >
> > Dear Reviewer YdfR,
> >
> > Thank you for taking the time and effort to review our work.
> >
> > We invite you to review our responses to ensure that we have addressed all your concerns. If there are additional experiments or analyses you would like us to conduct, please let us know at your earliest convenience to allow sufficient time for us to complete them.
> >
> > Should you have any further questions, please feel free to reach out. If you find our responses satisfactory and believe our revisions have strengthened the paper, we kindly ask you to consider raising your evaluation and recommending the paper for acceptance.
> >
> > Best regards,
> > Authors of Submission11170

---

> > > ### Author Response · Authors · 2024-12-02
> > >
> > > Dear Reviewer YdfR,
> > >
> > > Thank you for your thoughtful review of our work. We have addressed your comments and invite you to review our responses. If additional clarifications or analyses are needed, please let us know.
> > >
> > > Best regards,
> > > Authors of Submission11170

---

### Author Response · Authors · 2024-11-23
**General Comment by Authors**

Thanks to all reviewers for their helpful comments. We have incorporated all suggestions, which have strengthened the paper substantially. We have submitted a revised version of the manuscript (with the revision shown in blue). Here are the main changes:

**Added experiments:**
- **Robustness of results.** During revision, we have significantly increased the number of runs used to probe object discriminability along the visual hierarchy (Fig.5).
- **Increase transformation factor data augmentation.** We added more runs with a transformation factor from 0.8 to 1 (Fig. 4).
- **Multi-objective optimization.** Included models jointly trained to predict single-neuron responses and correlation of the population (Supp. Fig. 8).
-  **Task-driven models.** We trained a linear readout to predict neural responses from the first three layer blocks of a frozen ResNet 50 trained on object recognition (Supp. Fig. 6).
- **Models trained independently for each area.** We trained the model to predict neurons only from a specific area (Supp. Fig. 7).
- **Trade-off between single-neuron responses and differentiability.** We now show the relationship between the power-law and the single-neuron performance (Supp. Fig. 9)

**Modified text:**
- Clarified the experimental results for Stringer and Froudarakis in the introduction (Sec. 1)
- Improved the description of the model architecture and CC_norm metric (Appendix A1)
- Clarified hierarchy in discriminability (Sec. 4)
- Expanded the results with the new experiments (Sec. 5 and Sec. 6)
- Improved limitations and discussion: limitations of the architecture, trade-off between single neuron and population-level properties (Sec. 7)

Additionally, we reply to each reviewer individually to address their questions and concerns.

---

### Author Response · Authors · 2024-11-25
**Message to reviewers**

As the review process approaches its final stages, we would like to thank all reviewers once again for their thoughtful feedback, which has greatly improved our work. We have carefully addressed all comments and made the necessary revisions. We kindly invite you to review our responses and updated manuscript to ensure we have addressed your concerns. If any additional clarifications or reasonable changes are required, please let us know, and we will do our best to accommodate them. We hope our revisions meet your expectations and sincerely appreciate your consideration in recommending our paper for acceptance.

---

### Meta-Review · Area_Chair_nWK7 · 2024-12-20

**Metareview:**

This paper examines the performance of “digital twin” models for neuroscience data, i.e. deep networks trained to predict the responses of neurons to different stimuli (here visual stimuli). The authors claim that digital twin models can capture single-neuron responses reasonably well while still failing to capture population-level properties of the representations (e.g. the geometry and differentiability of the representational manifolds). To support this claim, the authors examine the properties of a few different digital twin models tested on visual response datasets from mice, and show that they fail to capture population-level properties well. The authors then explore what design choices can be used to improve population-level matches, and find that choices that improve robustness and generalization (e.g. drop-out) can help in this regard.

The strengths of this paper are that it addresses an important topic for computational neuroscience, is clearly written, and uses appropriate metrics and data to support the claims. The weaknesses are that there were relatively few model architectures and datasets explored in the initial submission, and a lack of exploration of other potential solutions (e.g. multi-objective models).

Overall, the authors did a good job of addressing these weaknesses in the rebuttal, and the final average score was over 6. Based on this, a decision of accept (poster) was reached.

**Additional Comments On Reviewer Discussion:**

The authors worked hard to address the reviewers’ concerns. Two out of three of the reviewers didn’t reply to the authors’ rebuttal though, and the one that did bumped their score up to 8. Based in part on these considerations, and the AC’s assessment of the rebuttals, a decision to accept was most fair.

---

### Decision · Program_Chairs · 2025-01-22

Accept (Poster)